# Dichotomy of Feature Learning and Unlearning:
# Fast-Slow Analysis on Neural Networks with Stochastic Gradient Descent

**Shota Imai**[1]  **Sota Nishiyama**[1,2]  **Masaaki Imaizumi**[1,2,3]

## Abstract

The dynamics of gradient-based training in neural networks often exhibit nontrivial structures; hence, understanding them remains a central challenge in theoretical machine learning. In particular, the concept of *feature **un**learning*, in which a neural network progressively loses previously learned features over long training, has gained attention. In this study, we consider the infinite-width limit of a two-layer neural network trained with a large-batch stochastic gradient, then derive differential equations with different time scales, revealing the mechanism and conditions for feature unlearning to occur. Specifically, we utilize the *fast-slow dynamics*: while an alignment of first-layer weights develops rapidly, the second-layer weights develop slowly. The direction of the flow on a critical manifold, determined by the slow dynamics, decides whether feature unlearning occurs. We give numerical validation of the result and derive theoretical grounding and scaling laws for the feature unlearning. Our results yield the following insights: (i) the strength of the primary nonlinear term in the data induces the feature unlearning, and (ii) an initial scale of the second-layer weights mitigates the feature unlearning. Our result should be understood as a population loss of alignment rather than finite-sample overfitting. Technically, our analysis utilizes Tensor Programs and singular perturbation theory.

## 1. Introduction

**Background.** Understanding the dynamics of gradient-based training in neural networks is a central problem in modern machine learning. Beyond static characterizations such as loss landscapes or stationary points, it has become increasingly clear that many learning phenomena are inherently *dynamical*. Especially, in high-dimensional regimes, self-averaging often enables a drastic simplification: the learning dynamics can be described by a small number of macroscopic order parameters. Several theoretical frameworks make this reduction precise, i.e., the dynamical mean-field theory (Bordelon & Pehlevan, 2022; Celentano et al., 2021), the Tensor Programs (Yang, 2019a;b; 2020a; Yang & Littwin, 2021), and the generalized first-order method (Celentano et al., 2020). Research on the learning dynamics of high-dimensional neural networks is rapidly advancing.

Key discoveries from analyzing dynamics include *feature learning*, which refers to the process where shallow layers of neural networks learn the feature structures of data-generating models, which helps explain why multi-layer structures achieve better accuracy. These were analyzed in Ba et al. (2022); Damian et al. (2022); Moniri et al. (2024); Yang & Hu (2021), demonstrating that neural networks trained with appropriate design can achieve feature learning.

In contrast, *feature **un**learning* has been proposed as an important notion related to feature learning. Feature unlearning refers to the phenomenon where shallow layers of neural networks forget feature structures they have previously learned, and could serve as one theory explaining the mechanism of deep learning. Particularly, Montanari & Urbani (2025) studies a neural network updated by a gradient flow and identifies a pronounced separation of time scales in two-layer networks, together with regimes in which previously learned features are progressively forgotten. These results suggest that feature unlearning is not a pathological effect, but rather can be understood as a generic consequence of multiple time scales in high-dimensional training regimes. However, research on these important concepts is still in its infancy, since the analytical framework is currently limited to updates via gradient flow, hence its underlying *mechanism* remains incompletely understood.

**Motivation.** The purpose of this study is to investigate whether feature unlearning occurs in a more general neu-

---

[1]The University of Tokyo, Tokyo, Japan [2]RIKEN AIP, Tokyo, Japan [3]Kyoto University, Kyoto Japan. Correspondence to: Shota Imai <imai-shota0516@g.ecc.u-tokyo.ac.jp>.

*Proceedings of the $43^{rd}$ International Conference on Machine Learning*, Seoul, South Korea. PMLR 306, 2026. Copyright 2026 by the author(s).

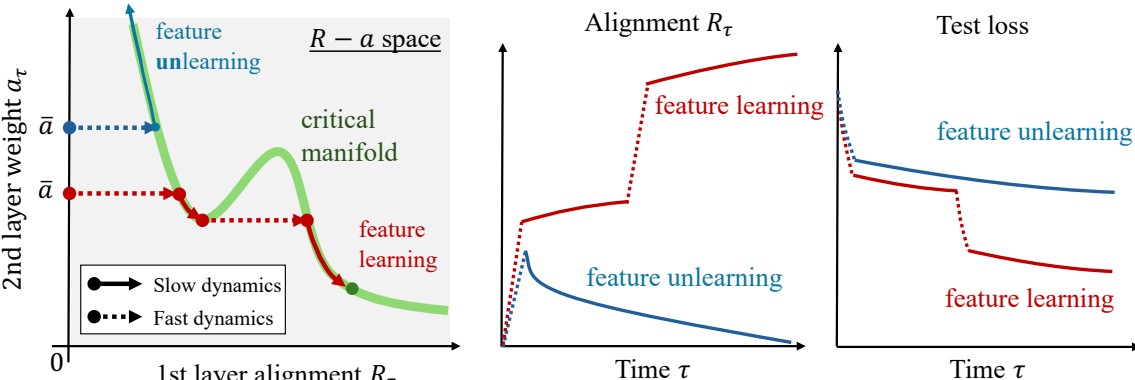

*Figure 1.* Fast-slow dynamics of first-layer alignment ($R_\tau$) and second-layer weights ($a_\tau$) in time $\tau$, explaining the evolution of alignment and test loss. In the space of $R_\tau$ and $a_\tau$, the $R-a$ space in the left panel, we find a *critical manifold* (green curve). Each trajectory starts from its initial point $(0, \bar{a})$ and, after reaching the manifold, slowly evolves along it. On all panels, the red trajectory represents feature learning, while the blue trajectory represents feature unlearning. The solid line represents slow dynamics, and the dashed line represents fast dynamics. In the feature learning case, test loss decreases like a staircase when $|R_\tau|$ increases via the fast dynamic. In the feature unlearning case, $|R_\tau|$ initially increases, then converges to zero due to the slow dynamics on the manifold.

ral network setting, namely, updates via stochastic gradient descent (SGD) in discrete time, and to clarify whether it also occurs there. Furthermore, it aims to reveal more rigorously the underlying principles of time scales by which feature unlearning occurs.

**Approach.** To this aim, we consider a neural network updated by one-pass SGD and derive a critical equation representing its dynamics. Starting from the setup with data generated from a single-index teacher model, we use the Tensor Programs and infinite-width limit, then derive a deterministic continuous-time differential equation to describe macroscopic variables of a neural network.

We then introduce an ansatz that reveals a separation of time scales among the variables of the derived system, casting the dynamics into a singularly perturbed problem. Although the ansatz is not assumed a priori at the level of SGD, numerical experiments show that it provides an accurate description of the observed dynamics. This reformulation allows us to derive a system to represent the long-time behavior of the macroscopic variables of the neural network and to isolate the reduced dynamics.

**Results and implications.** Our analysis shows that feature unlearning arises as a direct consequence of the reduced slow dynamics along a *critical manifold* induced from the derived system in the space of macroscopic variables. Specifically, in the infinite-width limit, trajectories of the macroscopic variables rapidly collapse onto the critical manifold and subsequently drift along it over much longer time scales, leading to a decay of feature alignment under explicit and verifiable conditions. We identify the structure of the reduced slow dynamics responsible for the onset of feature unlearning and derive the as-

sociated asymptotic scaling laws, thereby providing a concrete and quantitative dynamical mechanism for the phenomenon. Figure 1 outlines the mechanism.

We give additional insights into feature unlearning. Our result refers to the phenomenon that the alignment between the first-layer weights and the teacher direction is first acquired during an early phase of training and is later lost during long-time training. This phenomenon may also be interpreted as *feature unalignment*.

The main difference with Montanari & Urbani (2025) is the source and interpretation of the phenomenon. While Montanari & Urbani (2025) studies feature unlearning in relation to dataset-dependent overfitting for finite training data, we analyze population-level dynamics rather than a training-test gap since our setup is online SGD and a large-batch limit. Consequently, our result shows that even without finite-dataset memorization, the geometry of the population dynamics can drive the system toward vanishing alignment.

**Key novelty and approach.** The main contributions of this work are summarized as follows:

- *From discrete SGD to macroscopic dynamics*: Using the Tensor Programs framework, we derive a closed low-dimensional representation for online SGD and obtain, in the large-width limit, a deterministic ordinary differential equation (ODE) as a natural limit of the discrete algorithm.

- *Emergent fast-slow structure*: Numerical simulations of the limiting ODE reveal a clear separation of time scales, with fast convergence to a low-dimensional attracting set followed by slow evolution, referred to

as the fast-slow decomposition, justifying a singular-perturbation description. Based on the observation, we develop a new fast-slow system and analyze it theoretically.

- *Feature unlearning/learning as slow dynamics on the manifold*: Under the fast-slow structure, we show that feature unlearning/learning arises from the slow dynamics along the attracting critical manifold. Additionally, the staircase dynamics of test loss is described. Using the singular perturbation theory, we characterize the conditions for unlearning and derive its asymptotic scaling law.

- *Direct ODE analysis without ansatz*: In addition to the singular-perturbation formulation, we provide a direct nullcline-based analysis of the original two-dimensional ODE in the unlearning regime without the fast-slow ansatz.

## 1.1. Notation

Throughout the paper, $\| \cdot \|_2$ denotes the Euclidean norm. For a differentiable function $f : \mathbb{R} \to \mathbb{R}$, we write $f'$ and $f''$ for its first and second derivatives, respectively. For a multivariate function, $\nabla$ denotes the gradient. A function $f : \mathbb{R} \to \mathbb{R}$ is said to be *polynomially bounded* if there exist constants $C > 0$ and $k \geq 0$ such that $|f(x)| \leq C(1 + |x|^k)$, for all $x \in \mathbb{R}$. We use the standard Landau notations $O(\cdot)$, $o(\cdot)$, and $\Theta(\cdot)$ with their usual meanings. All random variables are defined on a common probability space, and convergence in probability is denoted by p-lim.

## 2. Setup

### 2.1. Online supervised learning

We study the supervised learning problem with an online learning setup. Suppose that there exists a random variable $(y, \boldsymbol{x}) \in \mathbb{R} \times \mathbb{R}^d$ with a dimension $d \in \mathbb{N}$, which is characterized by the following model, referred to as a teacher model, with a function $f_\star : \mathbb{R}^d \to \mathbb{R}$ as

$$\boldsymbol{x} \sim \mathcal{N}(0, I_d), \text{ and } y = f_\star(\boldsymbol{x}) + \varepsilon, \tag{1}$$

where $\varepsilon \sim \mathcal{N}(0, \sigma_\varepsilon^2)$ is an independent noise variable with its variance $\sigma_\varepsilon^2 > 0$. Here, $\boldsymbol{x}$ is a feature vector and $y$ is a response variable. The specific form of $f_\star$ will be formulated later. We assume that we observe data generated from this model (1) under the online learning setup described below.

We consider a two-layer neural network model as a model to be trained, referred to as a student model. Let $m \in \mathbb{N}$ be a width of the neural network and $\boldsymbol{W} = (\boldsymbol{w}_1, \ldots, \boldsymbol{w}_m) \in \mathbb{R}^{d \times m}$ and $\boldsymbol{a} = (a_1, \ldots, a_m)^\top \in \mathbb{R}^m$ be the first and second layer weights, respectively. Then, we study the neural

network $f(\cdot\,; \boldsymbol{a}, \boldsymbol{W}) : \mathbb{R}^d \to \mathbb{R}$ with an input $\boldsymbol{x} \in \mathbb{R}^d$ as

$$f(\boldsymbol{x}; \boldsymbol{a}, \boldsymbol{W}) = \frac{1}{m} \sum_{i=1}^m a_i \sigma(\langle \boldsymbol{w}_i, \boldsymbol{x} \rangle / \sqrt{d}). \tag{2}$$

We train the neural network model by online learning. In this setup, for each time $t \in \mathbb{N}$, we observe a set of $n$ pairs of responses and feature vectors $\{(y_i^t, \boldsymbol{x}_i^t)\}_{i=1}^n$, which are independent copies of $(y, \boldsymbol{x})$ from the model (1). We call the set a batch and $n$ the batch size. With the batch at time $t \in \mathbb{N}$, we define an empirical quadratic loss as

$$\mathcal{L}_t(\boldsymbol{a}, \boldsymbol{W}) = \frac{1}{2n} \sum_{i=1}^n \left( y_i^t - f(\boldsymbol{x}_i^t; \boldsymbol{a}, \boldsymbol{W}) \right)^2.$$

Then, we conduct one-pass stochastic gradient descent (SGD) to update the parameters of the neural network model. Specifically, with an initialization $\boldsymbol{a}^0$ and $\boldsymbol{W}^0$, the one-pass SGD generates sequences $\boldsymbol{a}^1, \boldsymbol{a}^2, \ldots$ and $\boldsymbol{W}^1, \boldsymbol{W}^2, \ldots$ by the following recursive form:

$$\boldsymbol{w}_i^{t+1} = \frac{\sqrt{d}}{\|\widetilde{\boldsymbol{w}}_i^{t+1}\|_2} \widetilde{\boldsymbol{w}}_i^{t+1}, \tag{3}$$

$$\widetilde{\boldsymbol{w}}_i^{t+1} = \boldsymbol{w}_i^t - \gamma d \nabla_{\boldsymbol{w}_i} \mathcal{L}_t(\boldsymbol{a}^t, \boldsymbol{W}^t),$$

for $i = 1, \ldots, m$ and

$$\boldsymbol{a}^{t+1} = \boldsymbol{a}^t - \gamma \nabla_{\boldsymbol{a}} \mathcal{L}_t(\boldsymbol{a}^t, \boldsymbol{W}^t), \tag{4}$$

where $\gamma > 0$ is a fixed learning rate. Here, we added a normalization step to the first-layer updates to guarantee that each $\boldsymbol{w}_i^t$ satisfies $\|\boldsymbol{w}_i^t\|_2 = \sqrt{d}$ ($i = 1, \ldots, m$) throughout training. Following Montanari & Urbani (2025), for theoretical convenience, we introduce a normalizing step to the first layer updates (3).

## 2.2. Conditions

Our theoretical analysis relies on several conditions. First, we provide the asymptotic settings for the batch size and the data dimension. This regime is common in theoretical analysis of neural networks, e.g., Celentano et al. (2020; 2021); Dandi et al. (2024).

**Assumption 1** (Proportionally high-dimensional regime)**.** *Both the batch size $n$ and the feature dimension $d$ diverge to infinity while preserving $n/d \to \delta$ with some $\delta \in (0, \infty)$.*

#### 2.2.1. TEACHER MODEL

We assume that the teacher model has the form of the single-index model. In particular, we introduce a specific form of $f_\star$ in (1):

**Assumption 2** (Single-index teacher)**.** *$f_\star : \mathbb{R}^d \to \mathbb{R}$ in (1) has the following form:*

$$f_\star(\boldsymbol{x}) = \sigma_\star \left( \langle \boldsymbol{w}_\star, \boldsymbol{x} \rangle / \sqrt{d} \right),$$

*with an unknown link function $\sigma_\star : \mathbb{R} \to \mathbb{R}$ and a teacher vector $\boldsymbol{w}_\star \sim \mathrm{Unif}(\mathbb{S}^{d-1}(\sqrt{d}))$, independently of $x$.*

The assumption of a single index as the teacher model is common in feature learning (Ba et al. (2022); Moniri et al. (2024)). The division by $\sqrt{d}$ in the input is necessary to maintain the variance of $\langle \boldsymbol{w}_\star, \boldsymbol{x} \rangle$ at a constant order even when $d$ diverges.

Next, we consider a property for the link function $\sigma_\star$ in Assumption 2. In preparation, we define the Hermite polynomial on $\mathbb{R}$. For $k \geq 1$, we define the $k$-th order Hermite polynomial as $H_0 = 1, H_1(x) = x, H_2(x) = x^2 - 1$, and generally

$$H_k(x) = (-1)^k \exp(x^2/2) \frac{d^k}{dx^k} \exp(-x^2/2), \quad x \in \mathbb{R},$$

which forms an orthogonal basis in the $L^2$-space. Then, we introduce the following condition:

**Assumption 3** (Degree of link function)**.** *A derivative $\sigma_\star'$ of $\sigma_\star$ exists and both $\sigma_\star$ and $\sigma_\star'$ are polynomially bounded. Also, we let $\sigma_\star$ have the following Hermite expansion in $L^2$ with $z \sim \mathcal{N}(0,1)$:*

$$\sigma_\star(\cdot) = \sum_{k=1}^{\infty} c_{\star,k} H_k(\cdot), \quad c_{\star,k} = \frac{1}{k!} \mathbb{E}[\sigma_\star(z) H_k(z)].$$

This assumption is commonly used in feature learning for neural networks employing single indices, e.g., Bietti et al. (2022); Damian et al. (2022); Ba et al. (2022); Cui et al. (2024); Dandi et al. (2024). Given $\sigma_\star(\cdot)$, we define a simple vector of the coefficients $c_\star := (c_{\star,1}, c_{\star,2}, ..., c_{\star,\bar{k}_\star})$ with $\bar{k}_\star := \max\{k : c_{\star,k} \neq 0\}$.

### 2.2.2. STUDENT MODEL AND TRAINING PROCESS

We introduce conditions for the neural network $f(\cdot; \boldsymbol{a}, \boldsymbol{W})$ and the training algorithm that are the subjects of training. The first concerns the activation function $\sigma : \mathbb{R} \to \mathbb{R}$:

**Assumption 4** (Degree of activation)**.** *Derivatives $\sigma'$ and $\sigma''$ of $\sigma$ exist, and $\sigma, \sigma', \sigma''$ are all polynomially bounded. Also, we let $\sigma$ have the following Hermite expansion in $L^2$ with $z \sim \mathcal{N}(0,1)$:*

$$\sigma(\cdot) = \sum_{k=1}^{\infty} c_k H_k(\cdot), \quad c_k = \frac{1}{k!} \mathbb{E}[\sigma(z) H_k(z)].$$

*We further assume that $c_{\star,1} c_1 > 0$ holds, and there exists some $k \geq 2$, such that $c_{\star,k} c_k \neq 0$.*

Given $\sigma(\cdot)$, we define a simple vector of the coefficients $c := (c_1, c_2, ..., c_{\bar{k}})$ with $\bar{k} := \max\{k : c_k \neq 0\}$.

This condition is analogous to the condition introduced for $\sigma_\star$ in Assumption 3. Such characterizations of link functions are also common in recent neural network theory (Ba

et al. (2022); Moniri et al. (2024)). Regarding condition $c_{\star,1} c_1 > 0$, analysis is similarly possible when $c_{\star,1} c_1$ is negative. However, since symmetry yields only similar results, we avoid unnecessary redundancy in the analysis by focusing on this case. The condition $c_{\star,k} c_k \neq 0$ for some $k \geq 2$ is formal and necessary for the analysis to properly handle the nonlinearity of the teacher and student models.

We introduce conditions for the initial values $\boldsymbol{a}^0$ and $\boldsymbol{W}^0$ for the SGD for online learning. Here, we utilize the symmetric initialization:

**Assumption 5** (Symmetric initialization)**.** *The initialization $\boldsymbol{a}^0$ and $\boldsymbol{W}^0$ are set as follows:*

$$a_i^0 = \bar{a} > 0, \quad \boldsymbol{w}_i^0 \overset{i.i.d.}{\sim} \mathcal{N}(0, \boldsymbol{I}_d), \quad (i = 1, \ldots, m).$$

This initialization is utilized by the seminal work Montanari & Urbani (2025) for the feature unlearning, where the second-layer weights are initialized to the same constant. This scheme reduces the number of substantial order parameters and helps to obtain effective low-dimensional expressions. It is also possible to analyze a case with $\bar{a} < 0$; we focus on the positive $\bar{a}$ to simplify our analysis.

## 3. Preparation: Reduction to fast-slow dynamics

We derive equations described by low-dimensional variables that illustrate how neural networks update through online learning. Specifically, our goal is to derive *singularly perturbed equations* capable of describing *fast-slow dynamics*, which reveals a separation of time scales in the resulting ODE system.

### 3.1. ODE of macroscopic variables

#### 3.1.1. MACROSCOPIC VARIABLES OF NEURAL NETWORKS

We introduce *macroscopic variables* to obtain a tractable and low-dimensional description of neural networks by SGD. For $t \in \mathbb{N}$ and $i, j = 1, \ldots, m$ with $i \neq j$, we define

$$R_i^m(t) := \underset{n,d \to \infty}{\mathrm{p\text{-}lim}} \frac{1}{d} \boldsymbol{w}_\star^\top \boldsymbol{w}_i^t, \quad \text{and} \quad a_i^m(t) := \underset{n,d \to \infty}{\mathrm{p\text{-}lim}} a_i^t \quad (5)$$

$R_i^m(t)$ measures the teacher-student alignment between the $i$-th weight vector and the teacher vector $\boldsymbol{w}_\star^\top$, and $a_i^m(t)$ is a scale of the $i$-th element of the second layer weight.

#### 3.1.2. ODE WITH INFINITE-WIDTH LIMIT

We next convert the difference equation to an ODE with continuous time by two approximations. The first is a mean-field limit $n, d \to \infty$; see Section C.2. The second approximation is the continuous-time limit of this recursion. In particular, we consider the infinite-width limit

$m \to \infty$ while the learning rate $\gamma$ is fixed. In this regime, the discrete dynamics becomes the following ODE system with continuous time $\tau = \gamma t/m \in \mathbb{R}_+$.

We introduce macroscopic variables $R_\tau$ and $a_\tau$ for $\tau \in \mathbb{R}_+$, which are continuous-time analogues of $R_i^m(t)$ and $a_i^m(t)$, respectively. Further, we define coefficient functions $S, T : \mathbb{R} \to \mathbb{R}$ as

$$S(z) = \sum_{k=1}^{\infty} k! c_{\star,k} c_k z^k, \quad T(z) = \sum_{k=1}^{\infty} k! c_k^2 z^{2k}.$$

Then, we define the following ODE:

> **ODE of macroscopic variables**: We define an ODE with $\{R_\tau, a_\tau\}_{\tau \in \mathbb{R}_+}$, as
>
> $$\frac{dR_\tau}{d\tau} = \underbrace{\frac{1}{2} a_\tau (1 - R_\tau^2)\{2S'(R_\tau) - a_\tau T'(R_\tau)\}}_{=:f(R_\tau, a_\tau)},$$
>
> $$\frac{da_\tau}{d\tau} = \underbrace{S(R_\tau) - a_\tau T(R_\tau)}_{=:g(R_\tau, a_\tau)},$$
> (6)
>
> with initialization $R_0 = 0$ and $a_0 = \bar{a} > 0$.

This equation describes the dynamics of neural networks with online SGD by a two-variate ODE. There are already several works on representing discrete online SGD via ODE (Goldt et al. (2019); Collins-Woodfin et al. (2024)). However, while previous studies derive ODE descriptions by taking a high-dimensional limit $d \to \infty$, we obtain the ODE by first considering the joint limit $n, d \to \infty$ and subsequently taking the infinite-width limit $m \to \infty$.

### 3.1.3. VALIDATION OF ODE

We prove the equivalence between the macroscopic variables of neural networks in (5) and the ODE (6) as follows.

**Proposition 1.** *Let* $R_{\tau,i}^m := R_i^m(\lfloor m\tau/\gamma \rfloor), a_{\tau,i}^m := a_i^m(\lfloor m\tau/\gamma \rfloor)$. *Then, for any finite* $\tau \geq 0$ *and any fixed* $i = 1, ..., m$, $R_\tau, a_\tau$ *solving the ODE* (6) *satisfies the following asymptotic equalities*

$$\lim_{m \to \infty} R_{\tau,i}^m = R_\tau, \quad and \quad \lim_{m \to \infty} a_{\tau,i}^m = a_\tau.$$

The proof is in Section C. We prove this equivalence by mediating the difference equation using the Tensor Program.

Also, in Section D, we derive the same ODE by an alternative approach of analyzing the population gradient.

## 3.2. Fast-slow reformulation

### 3.2.1. EMPIRICAL OBSERVATION OF MULTI TIME SCALE

We numerically solve the ODE (6) and study the dynamics of $(R_\tau, a_\tau)_{\tau \in \mathbb{R}_+}$ as shown in Figure 2. We observe a pronounced separation of time scales: $R_\tau$ rapidly changes over a short initial time interval, while $a_\tau$ remains nearly constant. After this fast transient, $R_\tau, a_\tau$ together evolve much more slowly. This behavior is consistently observed for different conditions.

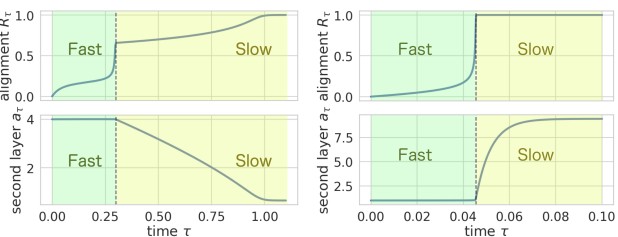

*Figure 2.* Multiple time scales appear in numerical simulations of (6). In the early stage of the dynamics, $R_\tau$ quickly moves away from $0$, while $a_\tau$ stays around the initial value $\bar{a}$. We set $\bar{k}_\star = \bar{k} = 5$ and $c = (1, 1, 1, 1, 1)$, and also set $c_\star = (1, -1, 1, -1, 1), \bar{a} = 4$ (left), and $c_\star = (2, 4, 6, 8, 10), \bar{a} = 1$ (right).

This separation of time scales can be explained as follows. First, since $R_\tau$ is zero at the initial stage, the update to $a_\tau$ is nearly zero, so only $R_\tau$ is updated initially. Second, when $R_\tau$ is small, we can observe that the eigenvalues of the Jacobian of the ODE concentrate on two peaks, with the peak in the direction of updating $R_\tau$ having a larger scale. In this situation, $R_\tau$ is updated preferentially. We will provide more quantitative discussion on this point in Section E.

### 3.2.2. FAST-SLOW ANSATZ

Motivated by numerical observations indicating that the dynamics of (6) exhibits a pronounced separation of time scales, we formalize this behavior by introducing two time-scales. Rather than postulating a small parameter at the level of the vector field, we identify representative fast and slow time-scales directly from the observed dynamics.

We introduce an ansatz for reducing (6) to a singularly perturbed system. In preparation, we define $\lambda_f(\tau), \lambda_s(\tau)$ as eigenvalues of a Jacobian of $F(R, a) := (f(R, a), g(R, a))^\top$ along a solution $(R_\tau, a_\tau)$. In the region analyzed in Appendix E, these eigenvalues are real and separated in magnitude; we denote them by $\lambda_f(\tau)$ and $\lambda_s(\tau)$, with $|\lambda_f(\tau)| \geq |\lambda_s(\tau)|$, corresponding to the fast and slow dynamics. Away from this region, since the Jacobian is not necessarily symmetric and the eigenvalues

can be complex, the same diagnostic may be replaced by the singular values, or by the magnitudes of the real parts when applicable. Also, $\boldsymbol{v}_f(\tau) \in \mathbb{R}^2$ denotes a normalized eigenvector corresponding to $\lambda_f(\tau)$.

**Ansatz** (Fast-slow dynamics). Given a time horizon $T$, the following hold:

1. (*Scale separation*) Time-averaged fast/slow eigenvalues, defined as

$$\Lambda_f := T^{-1} \int_0^T |\lambda_f(\tau)| d\tau \ \text{ and } \ \Lambda_s := T^{-1} \int_0^T |\lambda_s(\tau)| d\tau,$$

satisfy $\varepsilon := \Lambda_s/\Lambda_f \ll 1$.

2. (*Direction stability*) The eigenvector $\boldsymbol{v}_f(\tau)$ is well aligned to $\boldsymbol{e}_R = (1,0)^\top$ through time, that is, we have

$$T^{-1} \int_0^T \boldsymbol{v}_f(\tau)^\top \boldsymbol{e}_R d\tau \approx 1.$$

We can rigorously show the first point on the scale separation for small values of $|R|$: see Appendix E for details.

Such an ansatz is standard in the analysis of multi-scale dynamical systems and has been widely used across applied mathematics and theoretical physics. In the context of learning dynamics, closely related approaches have been employed (Jelbart et al., 2022; Montanari & Urbani, 2025; Berthier et al., 2025; Nishiyama & Imaizumi, 2025).

**Remark 1** (Necessity of the ansatz). We also note that the ansatz should be viewed as a singular-perturbation description of the observed dynamics, rather than as an additional assumption. We will also provide a direct nullcline-based analysis of the original ODE, which supports the same conclusion without relying solely on a global fast-slow ansatz.

Based on the fast-slow ansatz above, we rewrite (6) in terms of the slow time variable $\tau_s := \Lambda_s \tau$ and $\varepsilon > 0$.

---

**Singularly perturbed system**: We define $\{R^\varepsilon_{\tau_s}, a^\varepsilon_{\tau_s}\}_{\tau_s \in \mathbb{R}_+}$ by $R^\varepsilon_{\tau_s} := R_\tau$ and $a^\varepsilon_{\tau_s} := a_\tau$ as

$$\varepsilon \frac{dR^\varepsilon_{\tau_s}}{d\tau_s} = f(R^\varepsilon_{\tau_s}, a^\varepsilon_{\tau_s})/\Lambda_f =: \bar{f}(R^\varepsilon_{\tau_s}, a^\varepsilon_{\tau_s}),$$
$$\frac{da^\varepsilon_{\tau_s}}{d\tau_s} = g(R^\varepsilon_{\tau_s}, a^\varepsilon_{\tau_s})/\Lambda_s =: \bar{g}(R^\varepsilon_{\tau_s}, a^\varepsilon_{\tau_s}), \quad (7)$$

with initial conditions $R^\varepsilon_0 = 0$ and $a^\varepsilon_0 = \bar{a}$.

---

In this formulation, the variable $R$ evolves on the fast time scale $\Lambda_f \tau$, while $a$ evolves on the slow time scale $\Lambda_s \tau$. This regime differs from that considered in Berthier et al. (2025), where the second-layer weights are assumed to evolve on a faster time scale than the first-layer weights.

We view $\varepsilon$ as an independent small parameter and consider the limit $\varepsilon \to +0$. Following singular perturbation theory,

we define the following limiting values:

$$a^0_{\tau_s} := \lim_{\varepsilon \to +0} a^\varepsilon_{\tau_s}, \ \text{ and } \ R^0_{\tau_s} := \lim_{\varepsilon \to +0} R^\varepsilon_{\tau_s}.$$

# 4. Feature unlearning as slow flow

We present the feature unlearning phenomenon by numerical analysis on the derived models. While conceptual and theoretical analysis require the singularly perturbed system (7), we utilize the ODE (6) for numerical analysis because of its feasibility.

## 4.1. Feature learning and critical manifold

We first define the feature unlearning in the sense of the dynamics of the alignment $R^0_{\tau_s}$.

**Definition 1** (Feature unlearning). We say that a neural network system exhibits *feature unlearning*, if the variable for the alignment $\{R^0_{\tau_s}\}_{\tau_s \in \mathbb{R}_+}$ satisfies the following: there exists a constant $\bar{c} > 0$ and finite $\bar{\tau}$ such that we have

$$\max_{\tau_s \in (0, \bar{\tau})} |R^0_{\tau_s}| = \bar{c}, \ \text{ and } \ \lim_{\tau_s \to \infty} |R^0_{\tau_s}| = 0.$$

In contrast, when $\lim_{\tau_s \to \infty} |R^0_{\tau_s}|$ is lower bounded by a strictly positive constant, we say that the neural network achieves *feature learning*. This definition of feature unlearning implies that the first layer of a neural network aligns to the teacher vector $\boldsymbol{w}_\star$ as a feature at an early stage, and then the learned feature may be lost as training progresses. This definition conceptually follows Montanari & Urbani (2025).

Further, we formally define a manifold in the $R - a$ space, here termed a *critical manifold*, to which $\{R^0_{\tau_s}, a^0_{\tau_s}\}_{\tau_s \in \mathbb{R}_+}$ stays close in slow time.

**Definition 2** (Critical manifold). We define a critical manifold $\mathcal{S}$ by $\bar{f}(\cdot, \cdot)$ in the singularly perturbed system (7) as

$$\mathcal{S} := \{(R, a) \in [-1, 1] \times \mathbb{R} \mid \bar{f}(R, a) = 0\}.$$

As illustrated in Figure 1 and subsequent numerical analysis, $\mathcal{S}$ becomes a continuously smooth one-dimensional manifold. Note that $R^0_{\tau_s}$ takes a value only within $[-1, 1]$ by the form of ODE and the normalization of the online SGD (3), hence it is sufficient to consider $R \in [-1, 1]$.

## 4.2. Slow flow on the critical manifold

We analyze the dynamics of the macroscopic variables $(R_\tau, a_\tau)$ in the space of $R$ and $a$, i.e. an $R - a$ space. Numerically, we utilize the ODE (6) as a proxy of the singularly perturbed system (7), which is a common approach in singular perturbation theory (Hek, 2010) and its application to machine learning (Serino et al., 2025; Patsatzis et al., 2024b).

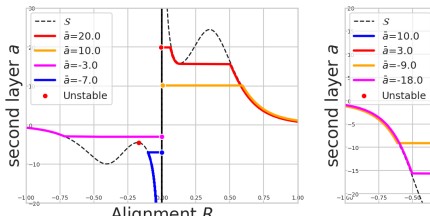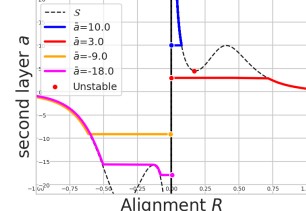

*Figure 3.* Trajectories of the model (6) on the $R - a$ space. The dots on the $a$-axis are the initial values $\bar{a}$. The red, yellow, pink trajectories show feature learning, and the blue trajectories show feature unlearning. We set $\bar{k}_\star = \bar{k} = 5$ and $c = (1,1,1,1,1)$, and also set $c_\star = (1,1,1,1,1)$ (left) or $c_\star = (1,-1,1,-1,1)$ (right).

The result, illustrated in Figure 3, reveals that there are two types of dynamics. In the fast time scale, the alignment $R_\tau$ evolves rapidly while $a_\tau$ remains effectively frozen, and trajectories are attracted to the critical manifold $\mathcal{S}$.

The trajectory first approaches the intersection between the horizontal line $a = \bar{a}$ and the stable branch

$$a = h(R), \quad \text{where} \quad h(R) := \frac{2S'(R)}{T'(R)}.$$

Consequently, the fast landing point is approximately $R_{\text{land}} = h^{-1}(\bar{a})$. After this landing, the direction of the slow flow is governed by the sign of

$$\alpha(R) := S(R)T'(R) - 2S'(R)T(R).$$

Thus, the initial second-layer scale $\bar{a}$ and the Hermite coefficients enter in two ways: they determine the landing point through $h$, and they determine the subsequent slow direction through $\alpha$.

Then, on longer time scales, the evolution is governed by a reduced slow flow along $\mathcal{S}$. The direction of the slow flow on $\mathcal{S}$ is determined by the link $\sigma_\star(\cdot)$ and the activation $\sigma(\cdot)$, and if there are unstable points on $\mathcal{S}$, the direction changes there. Even in dynamics on $\mathcal{S}$, some trajectories leave $\mathcal{S}$ and only $R_\tau$ evolves independently of $\mathcal{S}$. In such cases, the slow dynamics resumes upon reaching $\mathcal{S}$ again.

By further analysis on the longer time scale, we can observe two types of trajectories: **(I) one converges to a finite point** $\lim_{\tau \to \infty}(R_\tau, a_\tau) \to (R', a') \in \mathcal{S}$, and **(II) one diverges with zero alignment**, i.e. it behaves as $\lim_{\tau \to \infty}(R_\tau, a_\tau) \to (0, \pm\infty)$. Which trajectory appears depends on which point on the critical manifold $\mathcal{S}$ is reached on the fast time scale.

Consequently, the trajectory (II), which has diverging dynamics on $\mathcal{S}$, exhibits feature unlearning. Specifically, along certain branches, the reduced dynamics first drives $|R_\tau|$ away from zero and drives $R_\tau$ gradually toward zero afterwards. This behavior corresponds to a progressive loss of alignment $R_{\tau_s}^0$, and can be naturally interpreted as fea-

ture unlearning emerging from the slow drift along the attracting branch. We give more plots with different choices of $\sigma(\cdot)$ and $\sigma_\star(\cdot)$ in Section G.

### 4.3. Fast-slow flow and loss dynamics

We study the connection between the fast-slow dynamics in the $R - a$ space described above and the dynamics of other variables. Throughout this subsection, the loss is the population loss. Figure 4 compares a trajectory on the $R-a$ space with the corresponding transitions of the alignment $R_\tau$, the second-layer scale $a_\tau$, and the test loss.

In the initial fast dynamics where alignment $R_\tau$ increases from $0$, a rapid improvement in $R_\tau$ and a rapid decrease in loss occur. Subsequently, in the dynamics (I) where feature learning occurs, as $R_\tau$ increases slowly, the loss also gradually decreases. Furthermore, when the trajectory temporarily leaves the critical manifold $\mathcal{S}$ and evolves rapidly, the loss also decreases rapidly. In contrast, in the dynamics (II), where feature unlearning occurs, the trajectory diverges while $R_\tau$ decreases monotonically to zero. The loss may continue to decrease, but the limiting state has vanishing teacher alignment. Note that the trajectory is not lazy in the strict NTK sense, because $R_\tau$ first moves substantially away from its initialization before decaying.

## 5. Theoretical grounding

We derive a theoretical description of the feature unlearning based on the singularly perturbed system (7). This analysis mathematically supports the observation on the feature unlearning in Section 4.2.

### 5.1. Condition of feature unlearning

We provide a rigorous justification of feature unlearning in Definition 1. First, we impose an assumption on the relation between the link $\sigma_\star(\cdot)$ and the activation function $\sigma(\cdot)$:

**Assumption 6** (Redundant degree of polynomial of activation). *With $k_0 := \min\{k \geq 2 : c_{\star,k}c_k \neq 0\}$ and $k_1 = \min\{k \geq 2; c_k \neq 0\}$, one of the following holds:*

*(i)* $k_0 + 1 > 2k_1$,
*(ii)* $k_0 + 1 < 2k_1$ *and* $c_{\star,k_0}c_{k_0} < 0$,

Both of these conditions refer to a situation where the student model possesses low-order nonlinearities that are not present in the teacher model.

We next introduce an assumption on the initialization $\bar{a}$ through functions which may describe the dynamics on the critical manifold $\mathcal{S}$. Here, to simplify the analysis, we consider the case with $R > 0$.

**Assumption 7** (Initialization on $\bar{a}$). *We define the following values* $R_h = \min\{R \in (0,1) \mid h'(R) = 0\}$ *and* $R_\alpha =$

*Figure 4.* Simulated trajectories, alignments, second-layer weights, and losses of the model (6). We set $\bar{k}_\star = \bar{k} = 5$ and $c = (1, 1, 1, 1, 1), c_\star = (1, -1, 1, 1, 1)$, and $\bar{a} \in \{5, 9, 10\}$. We can observe that the learning dynamics proceeds differently for each case.

$\min\{R \in (0, 1) \mid \alpha(R) = 0\}$, *taking* $\min \emptyset = 1$, *and also define* $R^\star = \min\{R_h, R_\alpha\}$. *Then, we assume that there exists some* $R \in (0, R^\star)$ *such that* $\bar{a} = h(R)$ *holds.*

This assumption requires that the initial value $\bar{a}$ lies in a region that induces unlearning, i.e., divergence of the dynamics on $\mathcal{S}$. Here, $h(\cdot)$ is the parameterization of $a^0_{\tau_s}$ on $\mathcal{S}$ with respect to $R^0_{\tau_s}$, and $\alpha(\cdot)$ is a component of the intrinsic dynamics of $R^0_{\tau_s}$ on $\mathcal{S}$. These characterize the region of $(R, a) \in \mathcal{S}$ that moves toward divergence.

More precisely, the direction of the slow flow on $\mathcal{S}$ is determined by Assumption 7. We can observe that the direction of the slow dynamics on $\mathcal{S}$ is entirely determined by the sign of $\alpha(h^{-1}(a))$. Consequently, when $h^{-1}(\bar{a})$ crosses a root of $\alpha(R) = 0$ due to a change in the initial value $\bar{a}$, the direction of the slow flow is reversed. In particular, since Assumption 6 guarantees that $\alpha(R) > 0$ holds in a neighborhood of $R = +0$, feature unlearning occurs when $h^{-1}(\bar{a})$ is smaller than the smallest positive root of $\alpha(R) = 0$. Details are given in the reduced ODE (27) in Section F.1.

With these assumptions, we now state the theorem for feature unlearning:

**Theorem 2** (Feature unlearning). *Under Assumptions 3-7, we obtain*

$$\lim_{\tau_s \to \infty} R^0_{\tau_s} = 0, \text{ and } \lim_{\tau_s \to \infty} a^0_{\tau_s} = \infty.$$

*Furthermore, we have* $\lim_{\tau_s \to \infty} R^0_{\tau_s} a^0_{\tau_s} = c_{\star,1}/c_1$.

This theorem derives a sufficient condition for feature unlearning to occur. Specifically, under the assumptions imposed here, the divergence of $a^0_{\tau_s}$ and the vanishing of $R^0_{\tau_s}$, which represents alignment, indicate that the first layer weights lose the learned features, meaning learning occurs in the so-called lazy regime. The third limit of $R^0_{\tau_s} a^0_{\tau_s}$ provides additional information, that is, the rate of divergence of $a^0_{\tau_s}$ is $O((R^0_{\tau_s})^{-1})$.

This result can be regarded as a more precise description of the conditions under which feature unlearning occurs, as demonstrated by Montanari & Urbani (2025) under a similar setting.

**Remark 2** (Feature learning case). It is also possible to

derive a sufficient condition under which feature unlearning does not occur, i.e., feature learning occurs. Details will be provided in Section F.3.

### 5.2. Scaling law of feature unlearning

We derive a scaling law of the variables $R^0_{\tau_s}$ and $a^0_{\tau_s}$ in the feature unlearning case, which shows their convergence rate in terms of $\tau_s \to \infty$.

**Theorem 3** (Scaling law). *Under Assumptions 3-7, for each case of (i) and (ii) in Assumption 6, we obtain the following as* $\tau_s \to \infty$:

(i) $R^0_{\tau_s} = \Theta(\tau_s^{-1/(2k_1)})$, and $a^0_{\tau_s} = \Theta(\tau_s^{1/(2k_1)})$,
(ii) $R^0_{\tau_s} = \Theta(\tau_s^{-1/(k_0+1)})$, and $a^0_{\tau_s} = \Theta(\tau_s^{1/(k_0+1)})$,

These results imply that $k_0$ and $k_1$ defined in Assumption 6 are essential in determining the speed of convergence.

### 5.3. Feature unlearning without the fast-slow ansatz

We prove an analogue of Theorem 2 for the original ODE (6). We impose an additional assumption on the initialization $\bar{a}$.

**Assumption 8** (Initialization on $\bar{a}$). *We define* $H : (0, 1) \to \mathbb{R}$ *as* $H(R) = \frac{S(R)}{T(R)}$ *and define* $R_H = \min\{R \in (0, 1) \mid H'(R) = 0\}$. *Define* $R^{\star\star} = \min\{R_h, R_\alpha, R_H\}$. *Then, we assume that there exists some* $R \in (0, R^{\star\star})$ *such that* $\bar{a} = h(R)$ *holds.*

That is, in addition to Assumption 7 which assumes that the $R$-nullcline $a = h(R)$ is monotonically decreasing for $R < R^\star$, we assume that the $a$-nullcline $a = H(R)$ is also monotonically decreasing for $R < R^{\star\star}$.

We show feature unlearning for the original ODE (6).

**Theorem 4** (Feature unlearning). *Under Assumptions 3-8, we obtain*

$$\lim_{\tau \to \infty} R_\tau = 0, \text{ and } \lim_{\tau \to \infty} a_\tau = \infty.$$

Figure 5 illustrates the proof of Theorem 4. At the boundary of a region $K = \{(R, a) \mid \bar{a} \le a \le H(R), 0 \le R \le R^{\star\star}\}$ (the region surrounded by the dotted lines and the red curve), the vector field points inside $K$.

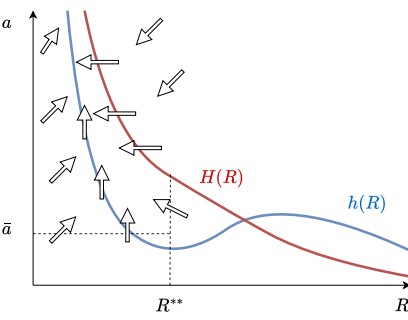

*Figure 5.* Illustration of the vector field by the ODE (6).

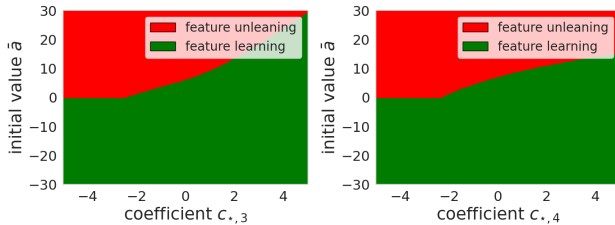

*Figure 6.* Phase maps for the feature unlearning by (6). We set $\bar{k}_\star = \bar{k} = 5$ and $c = (1,1,1,1,1)$, and also set $c_{\star,1} = c_{\star,4} = c_{\star,5} = 1, c_{\star,2} = -1$ (left), or $c_{\star,1} = c_{\star,3} = c_{\star,5} = 1, c_{\star,2} = -1$ (right).

## 6. Simulation

### 6.1. Phase map of feature unlearning

We perform numerical simulations of the ODE (6) for multiple choices of activation and link function coefficients $(c_k, c_{\star,k})$. Based on these simulations, we also construct a phase diagram summarizing whether unlearning occurs as a function of the coefficients and the initial value $\bar{a}$. Figure 6 shows the result. We see that, in this case, sign matching between the teacher/student coefficient is important for successful feature learning.

### 6.2. Scaling law of feature unlearning

We numerically investigate the convergence rates predicted by Theorem 3. By tracking the long-time behavior of $R_\tau$ and $a_\tau$ in the ODE (6), we find clear power-law regimes whose exponents agree with the theoretical scalings. These results provide quantitative confirmation of the scaling law for feature unlearning derived from the singular perturbation analysis. We observe that, in Figure 7, the log-log tail slopes of $R_\tau$ and $a_\tau$ of both settings approach the theoretical values $\pm 1/4$ and $\pm 1/3$, respectively.

### 6.3. Experiments with real neural networks and SGD

We report simulations of online SGD applied directly to real two-layer neural networks. Figure 8 shows the re-

sults. Across all tested configurations, we observe qualitative behaviors consistent with fast-slow dynamics, including a gradual decay of alignment with the teacher direction, accompanied by growth in the second-layer weights. While finite-width effects and stochastic fluctuations remain visible, these results suggest that the fast-slow mechanism predicted by the infinite-width theory persists, at least transiently, in realistic neural network settings.

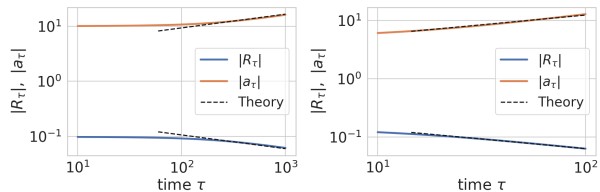

*Figure 7.* Numerical verification of the scaling law of Theorem 3. (Left) $\bar{k}_\star = \bar{k} = 7$, $c = (1,1,1,1,1,1,1), c_\star = (1,0,0,0,0,0,0.5), \bar{a} = 10$. This corresponds to the case (i) of Assumption 6; (Right) $\bar{k}_\star = \bar{k} = 3$, $c = (1,1,1), c_\star = (1,-1,1), \bar{a} = 5$. This corresponds to the case (ii) of Assumption 6.

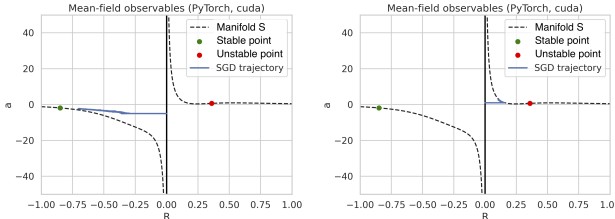

*Figure 8.* Numerical simulations of real neural networks up to $10^4$ iterations for $\bar{k}_\star = \bar{k} = 3$, $c = (1,1,1), c_\star = (1,-2,1)$, and learning rate $\gamma = 1$ with (Left) $n = d = 10^4, m = 500$; (Right) $n = d = 10^4, m = 10^3$. Stable/unstable points are where the flow direction on the manifold changes.

## 7. Conclusion

This paper investigates the learning dynamics of two-layer neural networks via SGD to elucidate the mechanism of feature unlearning. We clarified the relationship between the fast-slow dynamics of the macroscopic variables and the critical manifold, demonstrating that the slow flow on the manifold explains feature unlearning and staircase loss dynamics. A limitation is that since our result is essentially population-level, finding a connection to overfitting in finite samples remains a challenge for the future.

## Impact Statement

This paper presents work whose goal is to advance the theory of machine learning. There are many potential societal consequences of our work, none of which we feel must be specifically highlighted here.

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

# A. Related works

## A.1. Feature learning/unlearning in two-layer neural networks

The dynamics of feature learning in two-layer neural networks has been studied extensively in teacher-student settings and high-dimensional regimes (Ba et al. (2022); Damian et al. (2022); Moniri et al. (2024); Yang & Hu (2021)). The phenomenon of feature unlearning, where alignment with previously learned features degrades over long training times, was recently identified in large two-layer neural networks by Montanari & Urbani (2025). Using the dynamical mean-field theory, these authors revealed a pronounced separation of time scales and showed that feature unlearning can occur even under full-batch gradient flow in the infinite-width limit.

## A.2. Tensor Programs and other theories for high-dimensional dynamics

Tensor Programs provide a constructive and algorithm-aware framework for deriving macroscopic descriptions of wide neural networks directly from their computational graphs and update rules. Originally developed to analyze forward-pass behavior and signal propagation in deep networks (Yang, 2019a;b), the framework was later extended to cover backpropagation, training dynamics, and more general program structures (Yang, 2020a;b; Yang & Littwin, 2021; Yang & Hu, 2021; Yang et al., 2021; Littwin & Yang, 2023; Yang et al., 2024). Recent work has analyzed discrete gradient-based training algorithms, yielding rigorous state-evolution results for stochastic gradient descent and related methods (Gerbelot et al., 2024; Dandi et al., 2024). In contrast to the dynamical mean-field theory, which typically postulates a closed-form dynamical description at the outset, Tensor Programs offers a systematic procedure for deriving such descriptions from the underlying algorithm, a perspective that is central to the present work.

Recent works have focused on the high-dimensional limit, where learning dynamics can be characterized through a small number of macroscopic order parameters. Within this framework, both gradient flow and stochastic gradient descent have been studied using tools from dynamical mean-field theory, revealing the dependence of learning behavior on initialization, width, and data statistics (Bordelon & Pehlevan (2022); Celentano et al. (2021); Dandi et al. (2024); Mignacco et al. (2020)). Another theory is the generalized first-order theory, which handles a general class of iterative algorithms with first-order gradients and derives a state evolution to represent limiting high-dimensional dynamics of the iterative algorithms. Celentano et al. (2020) developed the framework and its efficiency, Han (2025) studies its applicability to a wider class of models and data distributions, and Han & Imaizumi (2025) applied the theory to the dynamics of multi-layer neural networks.

## A.3. Singular perturbation theory

Singular perturbation theory is a general theory to analyze multi-scale phenomena in dynamical systems and has long been used in physics, chemistry, biology, and many others (Fenichel, 1979; Patsatzis et al., 2024a; Serino et al., 2025; Jelbart et al., 2022; Kojakhmetov et al., 2021; Tran & Woldegerima, 2025). Regarding the analysis for neural networks, Berthier et al. (2025) analyzed incremental and non-monotone learning dynamics by explicitly introducing a small parameter to utilize the singular perturbation theory. Montanari & Urbani (2025) applied the singular perturbation theory for two-layer neural networks with gradient flow and analyzed the feature unlearning phenomenon as described above. Nishiyama & Imaizumi (2025) studied a diagonal linear neural network using singular perturbation theory and developed a precise dynamical analysis of the network training.

# B. Brief overview of Tensor Programs

## B.1. Program notation

Fix a parameter $n$. A $\textsc{NetSor}^\top$ program is a finite straight-line program whose vector variables lie in $\mathbb{R}^n$. The program is generated from an initial set of random vectors

$$\mathcal{V} = \{v^1, \dots, v^r\} \subset \mathbb{R}^n$$

and a set of Gaussian random matrices

$$\mathcal{W} = \{W^1, \dots, W^m\}, \qquad W^i \in \mathbb{R}^{n \times n}.$$

Each matrix $W \in \mathcal{W}$ has entries

$$W_{\alpha\beta} \sim \mathcal{N}\left(0, \frac{\sigma_W^2}{n}\right), \qquad 1 \le \alpha, \beta \le n,$$

independently over $\alpha, \beta$ and independently across distinct matrices $W$.

The initial vectors are sampled coordinatewise i.i.d. from a fixed finite-dimensional Gaussian law:

$$\left(v_\alpha^1, \ldots, v_\alpha^r\right) \overset{\text{i.i.d.}}{\sim} Z^{\mathcal{V}}, \qquad \alpha = 1, \ldots, n.$$

The program uses the following two instructions.

**Nonlinearity.** For $x^1, \ldots, x^k \in \mathbb{R}^n$ and $\phi : \mathbb{R}^k \to \mathbb{R}$, define

$$h := \phi(x^1, \ldots, x^k), \qquad h_\alpha = \phi(x_\alpha^1, \ldots, x_\alpha^k).$$

**Matrix multiplication.** For $W \in \mathcal{W}$ and $x \in \mathbb{R}^n$, define either

$$h := Wx \qquad \text{or} \qquad h := W^\top x.$$

It is convenient to write

$$W^+ := W, \qquad W^- := W^\top, \qquad \varepsilon \in \{+, -\},$$

so that a matrix multiplication has the form

$$h := W^\varepsilon x.$$

We also write

$$(W^\varepsilon)^\top = W^{-\varepsilon}.$$

## B.2. Symbolic $Z$-variables

To every vector variable $h$ appearing in the program, we associate a scalar random variable $Z^h$, which is intended to describe the limiting law of a typical coordinate $h_\alpha$. It is defined recursively from the program.

**Initial vectors.** For $v^i \in \mathcal{V}$, define

$$Z^{v^i} := (Z^{\mathcal{V}})_i.$$

Equivalently,

$$(Z^{v^1}, \ldots, Z^{v^r}) \sim Z^{\mathcal{V}}.$$

For initial vectors, set

$$\widehat{Z}^{v^i} := Z^{v^i}, \qquad \dot{Z}^{v^i} := 0.$$

**Nonlinearities.** If

$$h = \phi(x^1, \ldots, x^k),$$

then

$$Z^h := \phi(Z^{x^1}, \ldots, Z^{x^k}).$$

**Matrix multiplications.** Let

$$h = W^\varepsilon x, \qquad \varepsilon \in \{+, -\}.$$

Then define

$$Z^h = Z^{W^\varepsilon x} := \widehat{Z}^{W^\varepsilon x} + \dot{Z}^{W^\varepsilon x}.$$

The term $\widehat{Z}^{W^\varepsilon x}$ is the Gaussian part, and $\dot{Z}^{W^\varepsilon x}$ is the correction term.

### B.3. The Gaussian part

For each oriented matrix symbol $W^\varepsilon$, define

$$\mathcal{V}_{W^\varepsilon} := \left\{\, W^\varepsilon x \ : \ W^\varepsilon x \text{ appears in the program} \,\right\}.$$

The family

$$\left\{ \widehat{Z}^{W^\varepsilon x} : W^\varepsilon x \in \mathcal{V}_{W^\varepsilon} \right\}$$

is jointly Gaussian with mean zero and covariance

$$\mathrm{Cov}\left( \widehat{Z}^{W^\varepsilon x}, \widehat{Z}^{W^\varepsilon y} \right) = \sigma_W^2 \, \mathbb{E}\left[ Z^x Z^y \right].$$

Thus, for the same oriented matrix symbol $W^\varepsilon$,

$$\widehat{Z}^{W^\varepsilon x} \quad \text{and} \quad \widehat{Z}^{W^\varepsilon y}$$

are correlated exactly through the limiting inner product of $x$ and $y$.

If $W^\varepsilon \neq \overline{W}^\delta$ as oriented matrix symbols, then the corresponding Gaussian families are independent:

$$\left\{ \widehat{Z}^{W^\varepsilon x} : W^\varepsilon x \in \mathcal{V}_{W^\varepsilon} \right\} \perp \left\{ \widehat{Z}^{\overline{W}^\delta y} : \overline{W}^\delta y \in \mathcal{V}_{\overline{W}^\delta} \right\}.$$

In particular, the Gaussian symbols generated by $W$ and by $W^\top$ are treated as independent; their dependence is accounted for only through the correction term $\dot{Z}$.

### B.4. The correction term

The correction term records the dependence created when both $W$ and $W^\top$ occur in the same program. For

$$h = W^\varepsilon x,$$

define

$$\dot{Z}^{W^\varepsilon x} := \sigma_W^2 \sum_{\substack{y:\, W^{-\varepsilon} y \in \mathcal{V}_{W^{-\varepsilon}} \\ \widehat{Z}^{W^{-\varepsilon} y} \text{ appears in } Z^x}} Z^y \, \mathbb{E}\left[ \frac{\partial Z^x}{\partial \widehat{Z}^{W^{-\varepsilon} y}} \right].$$

Equivalently,

$$Z^{W^\varepsilon x} = \widehat{Z}^{W^\varepsilon x} + \sigma_W^2 \sum_{y:\, \widehat{Z}^{W^{-\varepsilon} y} \prec Z^x} Z^y \, \mathbb{E}\left[ \frac{\partial Z^x}{\partial \widehat{Z}^{W^{-\varepsilon} y}} \right],$$

where

$$\widehat{Z}^{W^{-\varepsilon} y} \prec Z^x$$

means that the symbolic expression defining $Z^x$ depends on the Gaussian symbol $\widehat{Z}^{W^{-\varepsilon} y}$.

Thus, in the special case where $x$ does not depend on any vector of the form $W^{-\varepsilon} y$, the correction vanishes:

$$\dot{Z}^{W^\varepsilon x} = 0, \qquad Z^{W^\varepsilon x} = \widehat{Z}^{W^\varepsilon x}.$$

### B.5. Interpretation of the partial derivative

For every program vector $x$, the symbolic variable $Z^x$ can be regarded as a deterministic function of finitely many Gaussian symbols:

$$Z^x = F_x\left( \widehat{Z}^{u^1}, \dots, \widehat{Z}^{u^q} \right),$$

where each $u^j$ is either an initial vector or a vector introduced by a matrix multiplication. Hence

$$\frac{\partial Z^x}{\partial \widehat{Z}^{W^{-\varepsilon} y}}$$

means the corresponding symbolic partial derivative of this function $F_x : \mathbb{R}^q \to \mathbb{R}$. If $\widehat{Z}^{W^{-\varepsilon} y}$ does not occur among the arguments of $F_x$, then

$$\frac{\partial Z^x}{\partial \widehat{Z}^{W^{-\varepsilon} y}} := 0.$$

**B.6. Example:** $A^\top A v$

Let

$$A_{\alpha\beta} \sim \mathcal{N}\left(0, \frac{1}{n}\right), \qquad v_\alpha \sim \mathcal{N}(0,1),$$

and consider the program

$$y := Av, \qquad x := A^\top y.$$

First,

$$Z^y = Z^{Av} = \widehat{Z}^{Av}, \qquad \widehat{Z}^{Av} \sim \mathcal{N}(0,1).$$

For $x = A^\top y$, we have

$$Z^x = Z^{A^\top y} = \widehat{Z}^{A^\top y} + \dot{Z}^{A^\top y}.$$

Since $y = Av$, the symbolic variable $Z^y$ depends on $\widehat{Z}^{Av}$. Therefore,

$$\dot{Z}^{A^\top y} = Z^v \, \mathbb{E}\left[\frac{\partial Z^y}{\partial \widehat{Z}^{Av}}\right] = Z^v \, \mathbb{E}[1] = Z^v.$$

Hence

$$Z^{A^\top A v} = \widehat{Z}^{A^\top y} + Z^v,$$

where

$$\widehat{Z}^{A^\top y} \sim \mathcal{N}(0,1), \qquad \widehat{Z}^{A^\top y} \perp Z^v.$$

Thus,

$$Z^{A^\top A v} \stackrel{d}{=} G + Z^v, \qquad G \sim \mathcal{N}(0,1), \qquad G \perp Z^v.$$

**B.7. Tensor Programs Master Theorem**

**Theorem 5** (NETSOR$^\top$ Master Theorem). *Fix a finite* NETSOR$^\top$ *program. Suppose that the initial matrices and initial vectors are sampled as above. Assume that all nonlinearities used in the program are polynomially bounded. Let*

$$h^1, \ldots, h^k$$

*be any finite collection of vector variables appearing in the program, and let*

$$\psi : \mathbb{R}^k \to \mathbb{R}$$

*be polynomially bounded. Then, as $n \to \infty$,*

$$\frac{1}{n} \sum_{\alpha=1}^{n} \psi\left(h^1_\alpha, \ldots, h^k_\alpha\right) \xrightarrow{\text{a.s.}} \mathbb{E}\left[\psi\left(Z^{h^1}, \ldots, Z^{h^k}\right)\right].$$

Here "polynomially bounded" means that there exist constants $C, c > 0$ and $p \geq 0$ such that

$$|\psi(z)| \leq C\|z\|^p + c \qquad \text{for all } z \in \mathbb{R}^k.$$

**B.8. Minimal summary**

A NETSOR$^\top$ Program consists of the recursive rules

$$
\boxed{
\begin{aligned}
&Z^v = (Z^{\mathcal{V}})_v \qquad v \in \mathcal{V}, \\[4pt]
&Z^{\phi(x^1, \ldots, x^k)} = \phi(Z^{x^1}, \ldots, Z^{x^k}), \\[4pt]
&Z^{W^\varepsilon x} = \widehat{Z}^{W^\varepsilon x} + \sigma_W^2 \sum_{\widehat{Z}^{W^{-\varepsilon} y} \prec Z^x} Z^y \, \mathbb{E}\left[\frac{\partial Z^x}{\partial \widehat{Z}^{W^{-\varepsilon} y}}\right], \\[4pt]
&\mathrm{Cov}\left(\widehat{Z}^{W^\varepsilon x}, \widehat{Z}^{W^\varepsilon y}\right) = \sigma_W^2 \mathbb{E}[Z^x Z^y].
\end{aligned}
}
$$

The Master Theorem states that these symbolic variables compute the almost-sure large $n$ limits of all empirical coordinate averages:

$$\frac{1}{n}\sum_{\alpha=1}^{n}\psi(h_\alpha^1,\dots,h_\alpha^k) \xrightarrow{\text{a.s.}} \mathbb{E}\left[\psi(Z^{h^1},\dots,Z^{h^k})\right].$$

In this paper, we consider the setting where the matrix and vectors have variable dimensions (not all equal to $n$), and we can easily extend this result to such cases, see Appendix F of (Yang, 2020b).

## C. Proof of Validation of ODE

### C.1. Difference equation of macroscopic variables for finite-width network

We derive the difference equation representing the dynamics of the macro variables. This equation is derived using the framework of Tensor Programs (Yang, 2019a; 2020a; Yang & Littwin, 2021), based on the neural network (2) and the online SGD update (3) and (4). In the following, we set $b_k := (k+1)\cdot(k+1)!$. For sequences or functions depending on a parameter $m$, we use the notation $O_m(\cdot)$.

In preparation, we define an additional macroscopic variable of neural networks trained by SGD:

$$Q_{i,j}^m(t) = \underset{n,d\to\infty}{\text{p-lim}}\, \frac{1}{d}\boldsymbol{w}_i^\top\boldsymbol{w}_j^t,\ \ i,j\in\{1,2,\dots,m\}, t=1,2,\dots.$$

$Q_{i,j}^m(t)$ corresponds to the overlap between $i$-th and $j$-th feature vectors. Note that for a case with $i=j$, $Q_{i,j}^m(t)=1$ holds because of the normalization step.

**Proposition 6** (Difference equation of macroscopic variable)**.** *There exists a sequence of macroscopic variables* $\{R^m(t), Q^m(t), a^m(t)\}_{t\in\mathbb{N}}$ *such that for any $i,j\in\{1,\dots,m\}$ we have*

$$R^m(t) = R_i^m(t), \quad Q^m(t) = Q_{i,j}^m(t)\ (i\neq j), \quad a^m(t) = a_i^m(t).$$

*Further, for any $t\in\mathbb{N}\cup\{0\}$, they satisfy the following recursive system*

$$
\begin{aligned}
R^m(t+1) &= R^m(t) + \gamma m^{-1}\Big\{ a^m(t)\big(1-R^m(t)^2\big)\sum_{k=0}^{\infty} b_k c_{\star,k+1}c_{k+1}R^m(t)^k \\
&\quad - a^m(t)^2 R^m(t)\big(1-Q^m(t)\big)\sum_{k=0}^{\infty} b_k c_{k+1}^2 Q^m(t)^k \Big\} + O_m(m^{-2}), \\
Q^m(t+1) &= Q^m(t) + \gamma m^{-1}\Big\{ 2a^m(t)R^m(t)\big(1-Q^m(t)\big)\sum_{k=0}^{\infty} b_k c_{\star,k+1}c_{k+1}R^m(t)^k \\
&\quad - 2a^m(t)^2 Q^m(t)\big(1-Q^m(t)\big)\sum_{k=0}^{\infty} b_k c_{k+1}^2 Q^m(t)^k \Big\} + O_m(m^{-2}), \\
a^m(t+1) &= a^m(t) + \gamma m^{-1}\Big\{ \sum_{k=1}^{\infty} k! c_{\star,k}c_k R^m(t)^k - a^m(t)\sum_{k=1}^{\infty} k! c_k^2 Q^m(t)^k \Big\} + O_m(m^{-2}),
\end{aligned}
\tag{8}
$$

*with initialization $R^m(0) = Q^m(0) = 0$ and $a^m(0) = \bar{a} \neq 0$.*

The proof and derivation process is given in Section C.2. Note that Assumptions 3 and 4 guarantee the absolute convergence of the infinite series that appear in the right-hand side of (8).

We have some implications of the difference equation. First, we can reduce the number of macroscopic variables, which consist of $O_m(m^2)$ components, that are intractable when $m\to\infty$, to only three variables under the symmetric initialization in Assumption 5. Details will be presented in Lemma 7 in Appendix C.2. Second, the macro variable following (8) is updated by balancing (i) the values determined by the interaction between $\sigma_\star$ and $\sigma$ through $c_{\star,k}$ and $c_k$, and (ii) the values determined solely by $\sigma$ through $c_k$.

## C.2. Derivation and validation of the discrete system

In this section, we rigorously derive recursive equations (8), and prove Lemma 7 at the same time. The actual derivation of the recursive dynamics is essentially based on Tensor Programs (Yang (2019a; 2020a;b); Yang & Hu (2021)). Following their notation of Tensor Programs, for a collection of potentially random $d$-dimensional vectors $x^1, x^2, ..., x^k \in \mathbb{R}^d$, we consider real-valued random variables $Z^{x^1}, ..., Z^{x^k}$ such that a distribution of elements of $x^j$ will be shown to converge to a distribution of $Z^{x^j}$ as $d \to \infty$, that is, it holds that

$$\frac{1}{d} \sum_{j=1}^{d} \psi(x_j^1, ..., x_j^k) \to \mathbb{E}[\psi(Z^{x^1}, ..., Z^{x^k})]$$

almost surely for every polynomially bounded $\psi : \mathbb{R}^k \to \mathbb{R}$. Further, we can decompose the random variable as $Z^{x^j} = \widehat{Z}^{x^j} + \dot{Z}^{x^j}$, where $\widehat{Z}^{x^j}$ and $\dot{Z}^{x^j}$ has specific formulation (see Box 2 in (Yang, 2020b). Detailed discussion and examples are given in (Yang, 2019a; 2020b).

In the following, we proceed by induction: assuming the statement of Lemma 7 and the equation (8) hold at the iteration $t$, we show they also hold for the iteration $t + 1$. Then we obtain that Lemma 7 and the recursive equation (8) hold.

**Preparation.** As a preliminary, we introduce several functions and constants for convenience. In Section C.2, we abbreviate notation by writing $R(t), Q(t), a(t)$ instead of $R^m(t), Q^m(t), a^m(t)$. First, for functions $a, b, c, d : \mathbb{R} \to \mathbb{R}$ that are integrable with respect to the Gaussian measure, define

$$
\begin{aligned}
&I_R(t; a, b) = \mathbb{E}[a(z_1)b(z_2)], \quad I_Q(t; a, b) = \mathbb{E}[a(z_3)b(z_4)], \\
&J_R(t; a, b, c) = \mathbb{E}[a(z_5)b(z_6)c(z_7)], \quad J_Q(t; a, b, c) = \mathbb{E}[a(z_8)b(z_9)c(z_{10})], \\
&K_R(t; a, b, c, d) = \mathbb{E}[a(z_{11})b(z_{12})c(z_{13})d(z_{14}))], \quad K_Q(t; a, b, c, d) = \mathbb{E}[a(z_{15})b(z_{16})c(z_{17})d(z_{18}))],
\end{aligned}
$$

where

$$
\begin{pmatrix} z_1(t) \\ z_2(t) \end{pmatrix} \sim GP\left(\mathbf{0}, \begin{pmatrix} 1 & R(t) \\ R(t) & 1 \end{pmatrix}\right), \quad \begin{pmatrix} z_3(t) \\ z_4(t) \end{pmatrix} \sim GP\left(\mathbf{0}, \begin{pmatrix} 1 & Q(t) \\ Q(t) & 1 \end{pmatrix}\right),
$$

$$
\begin{pmatrix} z_5(t) \\ z_6(t) \\ z_7(t) \end{pmatrix} \sim GP\left(\mathbf{0}, \begin{pmatrix} 1 & R(t) & R(t) \\ R(t) & 1 & Q(t) \\ R(t) & Q(t) & 1 \end{pmatrix}\right), \quad \begin{pmatrix} z_8(t) \\ z_9(t) \\ z_{10}(t) \end{pmatrix} \sim GP\left(\mathbf{0}, \begin{pmatrix} 1 & Q(t) & Q(t) \\ Q(t) & 1 & Q(t) \\ Q(t) & Q(t) & 1 \end{pmatrix}\right),
$$

$$
\begin{pmatrix} z_{11}(t) \\ z_{12}(t) \\ z_{13}(t) \\ z_{14}(t) \end{pmatrix} \sim GP\left(\mathbf{0}, \begin{pmatrix} 1 & R(t) & R(t) & R(t) \\ R(t) & 1 & Q(t) & Q(t) \\ R(t) & Q(t) & 1 & Q(t) \\ R(t) & Q(t) & Q(t) & 1 \end{pmatrix}\right), \quad \begin{pmatrix} z_{15}(t) \\ z_{16}(t) \\ z_{17}(t) \\ z_{18}(t) \end{pmatrix} \sim GP\left(\mathbf{0}, \begin{pmatrix} 1 & Q(t) & Q(t) & Q(t) \\ Q(t) & 1 & Q(t) & Q(t) \\ Q(t) & Q(t) & 1 & Q(t) \\ Q(t) & Q(t) & Q(t) & 1 \end{pmatrix}\right).
$$

Also, we further define the following constants:

$$s_1 = \mathbb{E}[\sigma'(z)^2], \quad s_2 = \mathbb{E}[\sigma(z)\sigma''(z)], \quad s_3 = \mathbb{E}[\sigma(z)^2\sigma'(z)^2], \quad s_4 = \mathbb{E}[\sigma(z)^2],$$

where $z \sim \mathcal{N}(0, 1)$. In what follows, we will derive the model (8), and prove Lemma 7 at the same time.

**Lemma 7** (Symmetricity of macroscopic variables)**.** *Under Assumption 5, we have $R_i^m(t) = R_j^m(t), Q_{i,j}^m(t) = Q_{i',j'}^m(t)$, and $a_i^m(t) = a_j^m(t)$ for any integer $t \geq 1$ and $i, j, i', j' \in \{1, \ldots, m\}$ that satisfy $i \neq j, i \neq j'$.*

We proceed with our proof inductively: let the model (8) and the statement of Lemma 7 hold for the $t$-th iteration, and prove the same statement for the $(t + 1)$-st iteration. We first calculate

$$
\begin{aligned}
\boldsymbol{G}_{\boldsymbol{w}}^t &= \nabla_{\boldsymbol{W}^t} \frac{1}{2n} \sum_{s=1}^{n} \left\{ y_s^t - \frac{1}{m} \sum_{i=1}^{m} a_i^t \sigma(\langle \boldsymbol{x}_s^t, \boldsymbol{w}_i^t \rangle / \sqrt{d}) \right\}^2 \\
&= \nabla_{\boldsymbol{W}^t} \frac{1}{2n} \left\| \boldsymbol{y}^t - \frac{1}{m} \sigma(\boldsymbol{X}^t \boldsymbol{W}^t) \boldsymbol{a}^t \right\|_2^2 \\
&= -\frac{1}{n} \cdot \frac{1}{m} \boldsymbol{X}^{t\top} \left\{ \left( \boldsymbol{y}^t \boldsymbol{a}^{t\top} - \frac{1}{m} \sigma(\boldsymbol{X}^t \boldsymbol{W}^t) \boldsymbol{a}^t \boldsymbol{a}^{t\top} \right) \odot \sigma'(\boldsymbol{X}^t \boldsymbol{W}^t) \right\} \in \mathbb{R}^{d \times m},
\end{aligned}
$$

where we defined $\boldsymbol{X}^t = (\boldsymbol{x}_1^t, \ldots, \boldsymbol{x}_n^t)^\top / \sqrt{d} \in \mathbb{R}^{n \times d}, \boldsymbol{a}^t = (a_1^t, \ldots, a_m^t)^\top \in \mathbb{R}^m$. We introduce

$$\boldsymbol{\ell}_t = \left( \boldsymbol{y}^t \boldsymbol{a}^{t\top} - \frac{1}{m} \sigma(\boldsymbol{X}^t \boldsymbol{W}^t) \boldsymbol{a}^t \boldsymbol{a}^{t\top} \right) \odot \sigma'(\boldsymbol{X}^t \boldsymbol{W}^t) =: (\boldsymbol{\ell}_{t,1}, \ldots, \boldsymbol{\ell}_{t,m}) \in \mathbb{R}^{d \times m}$$

and express

$$\boldsymbol{G}_{\boldsymbol{w}}^t = -\frac{1}{n} \cdot \frac{1}{m} \boldsymbol{X}^{t\top} \boldsymbol{\ell}_t =: (\boldsymbol{G}_{\boldsymbol{w},1}^t, \ldots, \boldsymbol{G}_{\boldsymbol{w},m}^t) \in \mathbb{R}^{d \times m}.$$

**About $R(t)$:** As the first step, we study the alignment term $R(t)$ through the analysis of $R_i(t)$. Since we have

$$R_i(t) = \mathbb{E}[Z^{\boldsymbol{w}^\star} Z^{\boldsymbol{w}_i^{t+1}}],$$

we mainly study the term $Z^{\boldsymbol{w}_i^{t+1}}$. To study this term, we recall the normalization step on the first layer:

$$\boldsymbol{w}_i^{t+1} = \frac{\sqrt{d}\widetilde{\boldsymbol{w}}_i^{t+1}}{\|\widetilde{\boldsymbol{w}}_i^{t+1}\|_2} = \frac{\widetilde{\boldsymbol{w}}_i^{t+1}}{\sqrt{\widetilde{\boldsymbol{w}}_i^{t+1\top} \widetilde{\boldsymbol{w}}_i^{t+1}/d}}, \quad \widetilde{\boldsymbol{w}}_i^{t+1} = \boldsymbol{w}_i^t - \gamma d \boldsymbol{G}_{\boldsymbol{w},i}^t,$$

Using the Tensor Programs formalism (Yang, 2019a), we obtain the form

$$Z^{\boldsymbol{w}_i^{t+1}} = Z^{\widetilde{\boldsymbol{w}}_i^{t+1}} / \sqrt{\mathbb{E}[(Z^{\widetilde{\boldsymbol{w}}_i^{t+1}})^2]}$$

$$= (Z^{\boldsymbol{w}_i^t} + \gamma Z^{-d\boldsymbol{G}_{\boldsymbol{w},i}^t}) / \sqrt{\mathbb{E}[(Z^{\boldsymbol{w}_i^t} + \gamma Z^{-d\boldsymbol{G}_{\boldsymbol{w},i}^t})^2]}. \tag{9}$$

Then, we will study the term $Z^{-d\boldsymbol{G}_{\boldsymbol{w},i}^t}$ and the expectation $\mathbb{E}[(Z^{\boldsymbol{w}_i^t} + \gamma Z^{-d\boldsymbol{G}_{\boldsymbol{w},i}^t})^2]$.

First, we directly study the term $Z^{-d\boldsymbol{G}_{\boldsymbol{w},i}^t}$. By the variable decomposition of the Tensor Programs, (e.g., Box 1 in (Yang, 2020b)), each element of $-d\boldsymbol{G}_{\boldsymbol{w},i}^t$ asymptotically follows $Z^{-d\boldsymbol{G}_{\boldsymbol{w},i}^t}$, which is decomposed as

$$Z^{-d\boldsymbol{G}_{\boldsymbol{w},i}^t} = \frac{1}{m\delta}\{\widehat{Z}^{\boldsymbol{X}^{t\top}\boldsymbol{\ell}_{t,i}} + \dot{Z}^{\boldsymbol{X}^{t\top}\boldsymbol{\ell}_{t,i}}\}, \tag{10}$$

where the variables $\widehat{Z}^{\boldsymbol{X}^{t\top}\boldsymbol{\ell}_{t,i}}$ and $\dot{Z}^{\boldsymbol{X}^{t\top}\boldsymbol{\ell}_{t,i}}$ follow Box 1 and Theorem 2.1 in (Yang, 2020b). About the term $\dot{Z}^{\boldsymbol{X}^{t\top}\boldsymbol{\ell}_{t,i}}$ in (10), since we assume the statement of Lemma 7 holds for the iteration $t$, we obtain the detailed form of $\dot{Z}^{\boldsymbol{X}^{t\top}\boldsymbol{\ell}_{t,i}}$ as

$$\dot{Z}^{\boldsymbol{X}^{t\top}\boldsymbol{\ell}_{t,i}}$$

$$= \delta a(t) \Big\{ \widehat{Z}^{\boldsymbol{w}^\star} \mathbb{E}[\sigma'_\star(\widehat{Z}^{\boldsymbol{X}^t\boldsymbol{w}^\star}) \sigma'(\widehat{Z}^{\boldsymbol{X}^t\boldsymbol{w}_i^t})]$$

$$+ Z^{\boldsymbol{w}_i^t} \mathbb{E}\Big[ -\frac{a(t)}{m} \sigma'(\widehat{Z}^{\boldsymbol{X}^t\boldsymbol{w}_i^t})^2 + \Big\{\sigma_\star(\widehat{Z}^{\boldsymbol{X}^t\boldsymbol{w}^\star}) + \widehat{Z}^{\boldsymbol{\varepsilon}^t} - \frac{a(t)}{m}\sum_{j=1}^m \sigma(\widehat{Z}^{\boldsymbol{X}^t\boldsymbol{w}_j^t})\Big\} \sigma''(\widehat{Z}^{\boldsymbol{X}^t\boldsymbol{w}_i^t})\Big]$$

$$+ \sum_{j \neq i} Z^{\boldsymbol{w}_j^t} \mathbb{E}\Big[ -\frac{a(t)}{m}\sigma'(\widehat{Z}^{\boldsymbol{X}^t\boldsymbol{w}_i^t})\sigma'(\widehat{Z}^{\boldsymbol{X}^t\boldsymbol{w}_j^t})\Big]\Big\}$$

$$= \delta a(t)\Big\{ \widehat{Z}^{\boldsymbol{w}^\star}\mathbb{E}[\sigma'_\star(\widehat{Z}^{\boldsymbol{X}^t\boldsymbol{w}^\star})\sigma'(\widehat{Z}^{\boldsymbol{X}^t\boldsymbol{w}_i^t})]$$

$$+ Z^{\boldsymbol{w}_i^t}\Big(\mathbb{E}[\sigma_\star(\widehat{Z}^{\boldsymbol{X}^t\boldsymbol{w}^\star})\sigma''(\widehat{Z}^{\boldsymbol{X}^t\boldsymbol{w}_i^t})] - \frac{a(t)}{m}\sum_{j\neq i}\mathbb{E}[\sigma(\widehat{Z}^{\boldsymbol{X}^t\boldsymbol{w}_j^t})\sigma''(\widehat{Z}^{\boldsymbol{X}^t\boldsymbol{w}_i^t})]$$

$$- \frac{a(t)}{m}\mathbb{E}[\sigma'(\widehat{Z}^{\boldsymbol{X}^t\boldsymbol{w}_i^t})^2] - \frac{a(t)}{m}\mathbb{E}[\sigma(\widehat{Z}^{\boldsymbol{X}^t\boldsymbol{w}_i^t})\sigma''(\widehat{Z}^{\boldsymbol{X}^t\boldsymbol{w}_i^t})]\Big)$$

$$- \frac{a(t)}{m}\sum_{j\neq i} Z^{\boldsymbol{w}_j^t}\mathbb{E}[\sigma'(\widehat{Z}^{\boldsymbol{X}^t\boldsymbol{w}_i^t})\sigma'(\widehat{Z}^{\boldsymbol{X}^t\boldsymbol{w}_j^t})]\Big\}$$

$$= \delta a(t)\Big\{ \widehat{Z}^{\boldsymbol{w}^\star} I_R(t; \sigma'_\star, \sigma') + Z^{\boldsymbol{w}_i^t}\Big(I_R(t; \sigma_\star, \sigma'') - a(t)\frac{m-1}{m}I_Q(t; \sigma, \sigma'') - \frac{a(t)}{m}s_1 - \frac{a(t)}{m}s_2\Big)$$

$$- \frac{a(t)}{m} \sum_{j \neq i} Z^{\boldsymbol{w}_j^t} I_Q(t; \sigma', \sigma') \Big\}.$$

To achieve this form, we utilize the form

$$Z^{\boldsymbol{\ell}_{t,i}} = a_i^t \Big\{ \sigma_\star(\widehat{Z}^{\boldsymbol{X}^t \boldsymbol{w}^\star}) + \widehat{Z}^{\boldsymbol{\varepsilon}^t} - \frac{1}{m} \sum_{j=1}^m a_j^t \sigma(\widehat{Z}^{\boldsymbol{X}^t \boldsymbol{w}_j^t}) \Big\} \sigma'(\widehat{Z}^{\boldsymbol{X}^t \boldsymbol{w}_i^t}). \tag{11}$$

Then, we can rewrite (10) as

$$
\begin{aligned}
mZ^{-d\boldsymbol{G}_{\boldsymbol{w},i}^t} \\
&= \frac{1}{\delta} (\widehat{Z}^{\boldsymbol{X}^{t\top} \boldsymbol{\ell}_{t,i}} + \dot{Z}^{\boldsymbol{X}^{t\top} \boldsymbol{\ell}_{t,i}}) \\
&= \frac{1}{\delta} \widehat{Z}^{\boldsymbol{X}^{t\top} \boldsymbol{\ell}_{t,i}} + a(t) \widehat{Z}^{\boldsymbol{w}^\star} I_R(t; \sigma_\star', \sigma') \\
&\quad + Z^{\boldsymbol{w}_i^t} \Big\{ a(t) I_R(t; \sigma_\star, \sigma'') - a(t)^2 I_Q(t; \sigma, \sigma'') + \frac{a(t)^2}{m} \big( I_Q(t; \sigma, \sigma'') - s_1 - s_2 \big) \\
&\quad - \frac{a(t)^2}{m} \sum_{j \neq i} Z^{\boldsymbol{w}_j^t} I_Q(t; \sigma', \sigma') \Big\}.
\end{aligned}
\tag{12}
$$

Second, we study the expectation $\mathbb{E}[(Z^{\boldsymbol{w}_i^t} + \gamma Z^{-d\boldsymbol{G}_{\boldsymbol{w},i}^t})^2]$ in (9), which includes the cross term $\frac{2\gamma}{m} \mathbb{E}[Z^{\boldsymbol{w}_i^t} m Z^{-d\boldsymbol{G}_{\boldsymbol{w},i}^t}]$ and the second moments $\mathbb{E}[(Z^{\boldsymbol{w}_i^{t+1}})^2]$ and $\mathbb{E}[(\frac{\gamma}{m} \cdot m Z^{-d\boldsymbol{G}_{\boldsymbol{w},i}^t})^2]$. In particular, we compute $\mathbb{E}[(Z^{\widetilde{\boldsymbol{w}}_i^{t+1}})^2] = \lim_{n,d\to\infty} \|\widetilde{\boldsymbol{w}}_i^{t+1}\|_2^2 / d$ following the relation $Z^{\widetilde{\boldsymbol{w}}_i^t} = Z^{\boldsymbol{w}_i^t} + \gamma/m \cdot m Z^{-d\boldsymbol{G}_{\boldsymbol{w},i}^t}$. About the cross term, we utilize (10) and obtain

$$
\begin{aligned}
\frac{2\gamma}{m} &\mathbb{E}[Z^{\boldsymbol{w}_i^t} m Z^{-d\boldsymbol{G}_{\boldsymbol{w},i}^t}] \\
&= \frac{2\gamma}{m} \Big\{ a(t) R(t) I_R(t; \sigma_\star', \sigma') + a(t) I_R(t; \sigma_\star, \sigma'') - a(t)^2 I_Q(t; \sigma, \sigma'') \\
&\quad + \frac{a(t)^2}{m} \big( I_Q(t; \sigma, \sigma'') - s_1 - s_2 \big) - a(t)^2 \frac{m-1}{m} Q(t) I_Q(t; \sigma', \sigma') \Big\} \\
&= \frac{2\gamma}{m} \Big\{ a(t) R(t) I_R(t; \sigma_\star', \sigma') + a(t) I_R(t; \sigma_\star, \sigma'') - a(t)^2 I_Q(t; \sigma, \sigma'') - a(t)^2 Q(t) I_Q(t; \sigma', \sigma') \Big\} \\
&\quad + \frac{2\gamma}{m^2} \Big\{ a(t)^2 \big( I_Q(t; \sigma, \sigma'') - s_1 - s_2 \big) + a(t)^2 Q(t) I_Q(t; \sigma', \sigma') \Big\} \\
&=: \frac{2\gamma}{m} A(t) + O_m(m^{-2}).
\end{aligned}
$$

Next, we calculate the second moments $\mathbb{E}[(\frac{\gamma}{m} \cdot m Z^{-d\boldsymbol{G}_{\boldsymbol{w},i}^t})^2]$ by utilizing the decomposition (10) as

$$\mathbb{E}\Big[\Big(\frac{\gamma}{m} \cdot m Z^{-d\boldsymbol{G}_{\boldsymbol{w},i}^t}\Big)^2\Big] = \frac{\gamma^2}{m^2} \mathbb{E}[(m Z^{-d\boldsymbol{G}_{\boldsymbol{w},i}^t})^2] = \frac{\gamma^2}{m^2} \Big\{ \mathbb{E}\Big[\Big(\frac{1}{\delta} \widehat{Z}^{\boldsymbol{X}^{t\top} \boldsymbol{\ell}_{t,i}}\Big)^2\Big] + \mathbb{E}\Big[\frac{1}{\delta}\Big(\dot{Z}^{\boldsymbol{X}^{t\top} \boldsymbol{\ell}_{t,i}}\Big)^2\Big] \Big\}.$$

With the relation $\mathbb{E}[(\frac{1}{\delta} \widehat{Z}^{\boldsymbol{X}^{t\top} \boldsymbol{\ell}_{t,i}})^2] = \frac{1}{\delta} \mathbb{E}[(Z^{\boldsymbol{\ell}_{t,i}})^2]$ and the form (11), it follows

$$
\begin{aligned}
\mathbb{E}[&(Z^{\boldsymbol{\ell}_{t,i}})^2] \\
&= a(t)^2 \mathbb{E}\Big[\Big\{ \sigma_\star(\widehat{Z}^{\boldsymbol{X}^t \boldsymbol{w}^\star}) + \widehat{Z}^{\epsilon^t} - \frac{a(t)}{m} \sum_{j=1}^m \sigma(\widehat{Z}^{\boldsymbol{X}^t w_j^t}) \Big\}^2 \sigma'(\widehat{Z}^{\boldsymbol{X}^t w_i^t})^2 \Big] \\
&= a(t)^2 \mathbb{E}[\sigma_\star(\widehat{Z}^{\boldsymbol{X}^t \boldsymbol{w}^\star})^2 \sigma'(\widehat{Z}^{\boldsymbol{X}^t w_i^t})^2] + \sigma_\varepsilon^2 a(t)^2 \mathbb{E}[\sigma'(\widehat{Z}^{\boldsymbol{X}^t w_i^t})^2] \\
&\quad - \frac{2a(t)^3}{m} \sum_{j \neq i} \mathbb{E}[\sigma_\star(\widehat{Z}^{\boldsymbol{X}^t \boldsymbol{w}^\star}) \sigma(\widehat{Z}^{\boldsymbol{X}^t w_j^t}) \sigma'(\widehat{Z}^{\boldsymbol{X}^t w_i^t})^2]
\end{aligned}
$$

$$-\frac{2a(t)^3}{m}\mathbb{E}[\sigma_\star(\widehat{Z}^{\boldsymbol{X}^t\boldsymbol{w}^\star})\sigma(\widehat{Z}^{\boldsymbol{X}^t w_i^t})\sigma'(\widehat{Z}^{\boldsymbol{X}^t w_i^t})^2]$$

$$+\frac{a(t)^4}{m^2}\sum_{\substack{j=1\\j\neq i}}^{m}\sum_{\substack{k=1\\k\neq i,k\neq j}}^{m}\mathbb{E}[\sigma(\widehat{Z}^{\boldsymbol{X}^t w_j^t})\sigma(\widehat{Z}^{\boldsymbol{X}^t w_k^t})\sigma'(\widehat{Z}^{\boldsymbol{X}^t w_i^t})^2]$$

$$+\frac{a(t)^4}{m^2}\sum_{\substack{j=1\\j\neq i}}^{m}\mathbb{E}[\sigma(\widehat{Z}^{\boldsymbol{X}^t w_j^t})^2\sigma'(\widehat{Z}^{\boldsymbol{X}^t w_i^t})^2]+\frac{a(t)^4}{m^2}\mathbb{E}[\sigma(\widehat{Z}^{\boldsymbol{X}^t w_i^t})^2\sigma'(\widehat{Z}^{\boldsymbol{X}^t w_i^t})^2]$$

$$=a(t)^2 I_R(t;\sigma_\star^2,\sigma'^2)+\sigma_\varepsilon^2 a(t)^2 s_1-2a(t)^3\frac{m-1}{m}J_R(t;\sigma_\star,\sigma,\sigma'^2)$$

$$-\frac{2a(t)^3}{m}I_R(t;\sigma_\star,\sigma\cdot\sigma'^2)+a(t)^4\frac{(m-1)(m-2)}{m^2}J_Q(t;\sigma,\sigma,\sigma'^2)$$

$$+a(t)^4\frac{m-1}{m^2}I_Q(t;\sigma^2,\sigma'^2)+\frac{a(t)^4}{m^2}s_3$$

$$=a(t)^2 I_R(t;\sigma_\star^2,\sigma'^2)+a(t)^2 s_1-2a(t)^3 J_R(t;\sigma_\star,\sigma,\sigma'^2)+a(t)^4 J_Q(t;\sigma,\sigma,\sigma'^2)$$

$$+\frac{1}{m}\Big\{2a(t)^3 J_R(t;\sigma_\star,\sigma,\sigma'^2)-2a(t)^3 I_R(t;\sigma_\star,\sigma\cdot\sigma'^2)$$

$$-3a(t)^4 J_Q(t;\sigma,\sigma,\sigma'^2)+a(t)^4 I_Q(t;\sigma^2,\sigma'^2)\Big\}$$

$$+\frac{1}{m^2}\Big\{2a(t)^4 J_Q(t;\sigma,\sigma,\sigma'^2)-a(t)^4 I_Q(t;\sigma^2,\sigma'^2)+a(t)^4 s_3\Big\}$$

$$=O_m(1). \tag{13}$$

Also, we obtain

$$\mathbb{E}\Big[\Big(\frac{1}{\delta}\dot{Z}^{\boldsymbol{X}^t\ell(\boldsymbol{X}^t\boldsymbol{w}^\star)_i}\Big)^2\Big]$$

$$=\mathbb{E}\Big[\Big\{a(t)\widehat{Z}^{\boldsymbol{w}^\star}I_R(t;\sigma_\star',\sigma')+Z^{w_i^t}\Big(a(t)I_R(t;\sigma_\star,\sigma'')-a(t)^2 I_Q(t;\sigma,\sigma'')$$

$$+\frac{a(t)^2}{m}(I_Q(t;\sigma,\sigma'')-s_1-s_2)\Big)-\frac{a(t)^2}{m}\sum_{j\neq i}Z^{\boldsymbol{w}_j}I_Q(t;\sigma',\sigma')\Big\}^2\Big]$$

$$=a(t)^2 I_R(t;\sigma_\star',\sigma')^2+\Big\{a(t)I_R(t;\sigma_\star,\sigma'')-a(t)^2 I_Q(t;\sigma,\sigma'')+\frac{a(t)^2}{m}\big(I_Q(t;\sigma,\sigma'')-s_1-s_2\big)\Big\}^2$$

$$+\frac{a(t)^4}{m^2}\mathbb{E}\Big[\Big(\sum_{j\neq i}Z^{\boldsymbol{w}_j}I_Q(t;\sigma',\sigma')\Big)^2\Big]$$

$$+2a(t)R(t)I_R(t;\sigma_\star',\sigma')\Big(a(t)I_R(t;\sigma_\star,\sigma'')-a(t)^2 I_Q(t;\sigma,\sigma'')+\frac{a(t)^2}{m}(I_Q(t;\sigma,\sigma'')-s_1-s_2)\Big)$$

$$-2a(t)^2\frac{m-1}{m}Q(t)I_Q(t;\sigma',\sigma')\Big(a(t)I_R(t;\sigma_\star,\sigma'')-a(t)^2 I_Q(t;\sigma,\sigma'')$$

$$+\frac{a(t)^2}{m}(I_Q(t;\sigma,\sigma'')-s_1-s_2)\Big)-2a(t)^3\frac{m-1}{m}R(t)I_R(t;\sigma_\star',\sigma')I_Q(t;\sigma',\sigma')$$

$$=a(t)^2 I_R(t;\sigma_\star',\sigma')^2+\Big(a(t)I_R(t;\sigma_\star,\sigma'')-a(t)^2 I_Q(t;\sigma,\sigma'')+\frac{a(t)^2}{m}(I_Q(t;\sigma,\sigma'')-s_1-s_2)^2\Big)$$

$$+a(t)^4 I_Q(t;\sigma',\sigma')^2\Big\{\frac{(m-1)(m-2)}{m^2}Q(t)+\frac{m-1}{m^2}\Big\}$$

$$+2a(t)R(t)I_R(t;\sigma_\star',\sigma')\Big(a(t)I_R(t;\sigma_\star,\sigma'')-a(t)^2 I_Q(t;\sigma,\sigma'')+\frac{a(t)^2}{m}(I_Q(t;\sigma,\sigma'')-s_1-s_2)\Big)$$

$$-2a(t)^2\frac{m-1}{m}Q(t)I_Q(t;\sigma',\sigma')\Big(a(t)I_R(t;\sigma_\star,\sigma'')-a(t)^2 I_Q(t;\sigma,\sigma'')$$

$$+\frac{a(t)^2}{m}(I_Q(t;\sigma,\sigma'')-s_1-s_2)\Big)$$

$$- 2a(t)^3 \frac{m-1}{m} R(t) I_R(t; \sigma'_\star, \sigma') I_Q(t; \sigma', \sigma')$$

$$= a(t)^2 I_R(t; \sigma'_\star, \sigma')^2 + \left\{ a(t) I_R(t; \sigma_\star, \sigma'') - a(t)^2 I_Q(t; \sigma, \sigma'') \right\}^2 + a(t)^4 I_Q(t; \sigma', \sigma')^2 Q(t)$$

$$+ 2a(t) R(t) I_R(t; \sigma'_\star, \sigma') \left\{ a(t) I_R(t; \sigma_\star, \sigma'') - a(t)^2 I_Q(t; \sigma, \sigma'') \right\}$$

$$- 2a(t)^2 Q(t) I_Q(t; \sigma', \sigma') \left\{ a(t) I_R(t; \sigma_\star, \sigma'') - a(t)^2 I_Q(t; \sigma, \sigma'') \right\}$$

$$- 2a(t)^3 R(t) I_R(t; \sigma'_\star, \sigma') I_Q(t; \sigma', \sigma')$$

$$+ m^{-1} \Big[ 2a(t)^2 \big( I_Q(t; \sigma, \sigma'') - s_1 - s_2 \big) \big( a(t) I_R(t; \sigma_\star, \sigma'') - a(t)^2 I_Q(t; \sigma, \sigma'') \big)$$

$$+ a(t)^4 I_Q(t; \sigma', \sigma')^2 \big( -3Q(t) + 1 \big) + 2a(t)^3 R(t) I_R(t; \sigma'_\star, \sigma') \big( I_Q(t; \sigma, \sigma'') - s_1 - s_2 \big)$$

$$- 2a(t)^2 Q(t) I_Q(t; \sigma', \sigma') \left\{ a(t)^2 \big( I_Q(t; \sigma, \sigma'') - s_1 - s_2 \big) - a(t) I_R(t; \sigma_\star, \sigma'') + a(t)^2 I_Q(t; \sigma, \sigma'') \right\}$$

$$+ 2a(t)^3 R(t) I_R(t; \sigma'_\star, \sigma') I_Q(t; \sigma', \sigma') \Big]$$

$$+ m^{-2} \left\{ a(t)^4 \big( I_Q(t; \sigma, \sigma'') - s_1 - s_2 \big)^2 + a(t)^4 I_Q(t; \sigma', \sigma')^2 \big( 2Q(t) - 1 \big) \right.$$

$$\left. + 2a(t)^4 Q(t) I_Q(t; \sigma', \sigma') \big( I_Q(t; \sigma, \sigma'') - s_1 - s_2 \big) \right\}$$

$$= O_m(1). \tag{14}$$

Combining the results (13) and (14) for $\mathbb{E}[(\frac{1}{\delta} \widehat{Z}^{\boldsymbol{X}^{t\top}} \boldsymbol{\ell}_{t,i})^2]$ and $\mathbb{E}[(\frac{1}{\delta} \dot{Z}^{\boldsymbol{X}^t \ell(\boldsymbol{X}^t \boldsymbol{w}^\star)_i})^2]$ with the relation (12), we get

$$\mathbb{E}\left[ \left( \frac{\gamma}{m} \cdot m Z^{-d\boldsymbol{G}^t_{\boldsymbol{w},i}} \right)^2 \right] = O_m(m^{-2}).$$

Thus, it follows that

$$\mathbb{E}[(Z^{\widetilde{\boldsymbol{w}}^{t+1}_i})^2] = \mathbb{E}[(Z^{\boldsymbol{w}^t_i} + \gamma Z^{-d\boldsymbol{G}^t_{\boldsymbol{w},i}})^2]$$

$$= \mathbb{E}[(Z^{\boldsymbol{w}^t_i} + \gamma m^{-1} \cdot m Z^{-d\boldsymbol{G}^t_{\boldsymbol{w},i}})^2]$$

$$= 1 + 2A(t)\gamma m^{-1} + O_m(m^{-2}),$$

then we evaluate the expectation term in (9) as

$$\{\mathbb{E}[(Z^{\boldsymbol{w}^t_i} + \gamma Z^{-d\boldsymbol{G}^t_{\boldsymbol{w},i}})^2]\}^{-1/2} = 1 - A(t)\gamma m^{-1} + O_m(m^{-2}). \tag{15}$$

Now, we are ready to study $R_i(t)$. By using (15), we update (15) as

$$Z^{\boldsymbol{w}^{t+1}_i} = Z^{\widetilde{\boldsymbol{w}}^{t+1}_i} / \sqrt{\mathbb{E}[(Z^{\widetilde{\boldsymbol{w}}^{t+1}_i})^2]}$$

$$= (Z^{\boldsymbol{w}^t_i} + \gamma Z^{-d\boldsymbol{G}^t_{\boldsymbol{w},i}}) / \sqrt{\mathbb{E}[(Z^{\boldsymbol{w}^t_i} + \gamma Z^{-d\boldsymbol{G}^t_{\boldsymbol{w},i}})^2]}$$

$$= (Z^{\boldsymbol{w}^t_i} + \gamma Z^{-d\boldsymbol{G}^t_{\boldsymbol{w},i}}) \times \{1 - A(t)\gamma m^{-1} + O_m(m^{-2})\}.$$

Therefore, by multiplying $Z^{\boldsymbol{w}^\star}$ on both sides and taking expectation, we derive

$$R_i(t+1) = \big( R(t) + \gamma \mathbb{E}[Z^{\boldsymbol{w}^\star} Z^{-d\boldsymbol{G}^t_{\boldsymbol{w},i}}] \big) \times \{1 - A(t)\gamma m^{-1} + O_m(m^{-2})\}$$

$$= \big( R(t) + \gamma m^{-1} \mathbb{E}[Z^{\boldsymbol{w}^\star} m Z^{-d\boldsymbol{G}^t_{\boldsymbol{w},i}}] \big) \times \{1 - A(t)\gamma m^{-1} + O_m(m^{-2})\}.$$

The appearing cross term is also evaluated using (12) as

$$\mathbb{E}[Z^{\boldsymbol{w}^\star} m Z^{-d\boldsymbol{G}^t_{\boldsymbol{w},i}}]$$

$$
\begin{aligned}
&= a(t)I_R(t;\sigma_\star',\sigma') + R(t)\Big\{a(t)I_R(t;\sigma_\star,\sigma'') - a(t)^2 I_Q(t;\sigma,\sigma'') + \frac{a(t)^2}{m}\big(I_Q(t;\sigma,\sigma'') - s_1 - s_2\big)\Big\} \\
&\quad - a(t)^2 \frac{m-1}{m} R(t) I_Q(t;\sigma',\sigma') \\
&= a(t)I_R(t;\sigma_\star',\sigma') + R(t)\Big\{a(t)I_R(t;\sigma_\star,\sigma'') - a(t)^2 I_Q(t;\sigma,\sigma'')\Big\} - a(t)^2 R(t) I_Q(t;\sigma',\sigma') \\
&\quad + \frac{1}{m}\Big\{a(t)^2 R(t)\big(I_Q(t;\sigma,\sigma'') - s_1 - s_2\big) + a(t)^2 R(t) I_Q(t;\sigma',\sigma')\Big\} \\
&=: B(t) + O_m(m^{-1}),
\end{aligned}
$$

and we have

$$
\begin{aligned}
R_i(t+1) &= \{R(t) + B(t)m^{-1} + O_m(m^{-2})\} \times \{1 - A(t)\gamma m^{-1} + O_m(m^{-2})\} \\
&= R(t) + \gamma m^{-1}\{-R(t)A(t) + B(t)\} + O_m(m^{-2}),
\end{aligned}
$$

where the right-hand side does not depend on $i$, so we can simply write $R(t+1)$, instead of $R_i(t+1)$. Here, we finally obtain

$$
R(t+1) = R(t) + \gamma m^{-1}\{-R(t)A(t) + B(t)\} + O_m(m^{-2}),
$$

where $-R(t)A(t) + B(t)$ is equal to the expression of the model (8).

**About $Q(t)$:** The derivation of the equation for $Q(t)$ proceeds quite similarly. Just like the above, we obtain

$$
\begin{aligned}
&Q_{i,j}(t+1) \\
&= \Big\{Q(t) + \gamma m^{-1}\mathbb{E}[Z^{\boldsymbol{w}_j^t} m Z^{-d\boldsymbol{G}_{\boldsymbol{w},i}^t}] + \gamma m^{-1}\mathbb{E}[Z^{\boldsymbol{w}_i^t} m Z^{-d\boldsymbol{G}_{\boldsymbol{w},j}^t}] + \gamma^2 m^{-2}\mathbb{E}[m Z^{-d\boldsymbol{G}_{\boldsymbol{w},i}^t} m Z^{-d\boldsymbol{G}_{\boldsymbol{w},j}^t}]\Big\} \\
&\quad \times \{1 - A(t)\gamma m^{-1} + O_m(m^{-2})\}^2 \qquad (i \neq j),
\end{aligned}
$$

where we can easily check $\mathbb{E}[Z^{\boldsymbol{w}_j^t} m Z^{-d\boldsymbol{G}_i^t}] = \mathbb{E}[Z^{\boldsymbol{w}_i^t} m Z^{-d\boldsymbol{G}_j^t}]$. We obtain

$$
\begin{aligned}
&\mathbb{E}[Z^{\boldsymbol{w}_j^t} m Z^{-d\boldsymbol{G}_i^t}] \\
&= a(t)R(t)I_R(t;\sigma_\star',\sigma') \\
&\quad + Q(t)\Big\{a(t)I_R(t;\sigma_\star,\sigma'') - a(t)^2 I_Q(t;\sigma,\sigma'') + \frac{a(t)^2}{m}\big(I_Q(t;\sigma,\sigma'') - s_1 - s_2\big)\Big\} \\
&\quad - a(t)^2 \frac{m-2}{m} Q(t) I_Q(t;\sigma',\sigma') - \frac{a(t)^2}{m} I_Q(t;\sigma',\sigma') \\
&= a(t)R(t)I_R(t;\sigma_\star',\sigma') + Q(t)\Big\{a(t)I_R(t;\sigma_\star,\sigma'') - a(t)^2 I_Q(t;\sigma,\sigma'')\Big\} - a(t)^2 Q(t)I_Q(t;\sigma',\sigma') \\
&\quad + \frac{1}{m}\Big\{a(t)^2 Q(t)\big(I_Q(t;\sigma,\sigma'') - s_1 - s_2\big) + 2a(t)^2 Q(t) I_Q(t;\sigma',\sigma') - a(t)^2 I_Q(t;\sigma',\sigma')\Big\} \\
&=: \frac{1}{2}C(t) + O_m(m^{-1}).
\end{aligned}
$$

The cross term can be decomposed as:

$$
\mathbb{E}[m Z^{-d\boldsymbol{G}_{\boldsymbol{w},i}^t} m Z^{-d\boldsymbol{G}_{\boldsymbol{w},j}^t}] = \frac{1}{\delta^2}\mathbb{E}[\widehat{Z}^{\boldsymbol{X}^{t\top}\boldsymbol{\ell}_{t,i}} \widehat{Z}^{\boldsymbol{X}^{t\top}\boldsymbol{\ell}_{t,j}}] + \frac{1}{\delta^2}\mathbb{E}[\dot{Z}^{\boldsymbol{X}^{t\top}\boldsymbol{\ell}_{t,i}} \dot{Z}^{\boldsymbol{X}^{t\top}\boldsymbol{\ell}_{t,j}}].
$$

The first term can be calculated as:

$$
\begin{aligned}
&\frac{1}{\delta^2}\mathbb{E}[\widehat{Z}^{\boldsymbol{X}^{t\top}\boldsymbol{\ell}_{t,i}} \widehat{Z}^{\boldsymbol{X}^{t\top}\boldsymbol{\ell}_{t,j}}] \\
&= \frac{a(t)^2}{\delta}\mathbb{E}\Big[\Big\{\sigma_\star(\widehat{Z}^{\boldsymbol{X}^t w^\star}) + \widehat{Z}^{\varepsilon^t} - \frac{a(t)}{m}\sum_{k=1}^m \sigma(\widehat{Z}^{\boldsymbol{X}^t w_k^t})\Big\}^2 \sigma'(\widehat{Z}^{\boldsymbol{X}^t w_i^t})\sigma'(\widehat{Z}^{\boldsymbol{X}^t w_j^t})\Big] \\
\\
&= \frac{a(t)^2}{\delta}\Big\{J_R(t;\sigma_\star^2,\sigma',\sigma') + I_Q(t;\sigma',\sigma') + a(t)^2 K_Q(t;\sigma,\sigma,\sigma',\sigma') - 2a(t)K_R(t;\sigma_\star,\sigma,\sigma',\sigma')\Big\}
\end{aligned}
$$

$$+ O_m(m^{-1}) = O_m(1).$$

Similarly, for the second term:

$$\mathbb{E}\left[\frac{1}{\delta}\dot{Z}\boldsymbol{X}^{t\top}\boldsymbol{\ell}_{t,i}\frac{1}{\delta}\dot{Z}\boldsymbol{X}^{t\top}\boldsymbol{\ell}_{t,j}\right]$$

$$= a(t)^2 I_R(t; \sigma'_\star, \sigma')^2$$

$$+ Q(t)\left\{a(t)I_R(t; \sigma_\star, \sigma'') - a(t)^2 I_Q(t; \sigma, \sigma'') + \frac{a(t)^2}{m}\big(I_Q(t; \sigma, \sigma'') - s_1 - s_2\big)\right\}^2$$

$$+ a(t)^4 Q(t) I_Q(t; \sigma', \sigma')^2$$

$$+ 2a(t)R(t)I_R(t; \sigma'_\star, \sigma')$$

$$\times \left\{a(t)I_R(t; \sigma_\star, \sigma'') - a(t)^2 I_Q(t; \sigma, \sigma'') + \frac{a(t)^2}{m}\big(I_Q(t; \sigma, \sigma'') - s_1 - s_2\big)\right\}$$

$$- 2a(t)^2 Q(t)I_Q(t; \sigma', \sigma')$$

$$\times \left\{a(t)I_R(t; \sigma_\star, \sigma'') - a(t)^2 I_Q(t; \sigma, \sigma'') + \frac{a(t)^2}{m}\big(I_Q(t; \sigma, \sigma'') - s_1 - s_2\big)\right\}$$

$$- 2a(t)^3 R(t)I_R(t; \sigma'_\star, \sigma')I_Q(t; \sigma', \sigma') + O_m(m^{-1})$$

$$= O_m(1).$$

Combining these results, the equation of $Q$ is reduced to the following:

$$Q_{i,j}(t+1) = \big\{Q(t) + C(t)\gamma m^{-1} + O_m(m^{-2})\big\} \times \big\{1 - A(t)\gamma m^{-1} + O_m(m^{-2})\big\}^2$$
$$= Q(t) + \big\{-2A(t)Q(t) + C(t)\big\}\gamma m^{-1} + O_m(m^{-2}).$$

We observe that $Q_{i,j}(t+1)$ does not depend on $i, j$, so we simply write it $Q(t+1)$, and obtain

$$Q(t+1) = Q(t) + \big\{-2A(t)Q(t) + C(t)\big\}\gamma m^{-1} + O_m(m^{-2}),$$

where $-2A(t)Q(t) + C(t)$ is the same as the form of (8).

**About** $a(t)$: As a final step, we derive the equation for $a(t)$. Because there is no normalization step for the second layer updates, the calculation is much simpler than that of $R(t)$ or $Q(t)$. Let $\boldsymbol{G}_{\boldsymbol{a}}^t = \nabla_{\boldsymbol{a}}\|\boldsymbol{y}^t - \frac{1}{m}\sum_{j=1}^m a_j^t\sigma(\boldsymbol{X}^t\boldsymbol{w}_j^t)\|_2^2$, then the second layer update proceeds as follows:

$$\boldsymbol{a}^{t+1} = \boldsymbol{a}^t - \gamma\boldsymbol{G}_{\boldsymbol{a}}^t$$
$$= \boldsymbol{a}^t - \gamma m^{-1} \cdot m\boldsymbol{G}_{\boldsymbol{a}}^t \in \mathbb{R}^m,$$

where $\boldsymbol{G}_{\boldsymbol{a}}^t := (\boldsymbol{G}_{\boldsymbol{a},1}^t, \ldots, \boldsymbol{G}_{\boldsymbol{a},m}^t)$. For each $\boldsymbol{G}_{\boldsymbol{a},i}^t$, one has

$$m\boldsymbol{G}_{\boldsymbol{a},i}^t = -\frac{1}{n}\sigma(\boldsymbol{X}^t\boldsymbol{w}_i^t)^\top\left\{\sigma_\star(\boldsymbol{X}^t\boldsymbol{w}^\star) + \boldsymbol{\varepsilon}^t - \frac{a(t)}{m}\sum_{j=1}^m \sigma(\boldsymbol{X}^t\boldsymbol{w}_j^t)\right\}$$

$$= -\frac{1}{n}\sigma(\boldsymbol{X}^t\boldsymbol{w}_i^t)^\top\sigma_\star(\boldsymbol{X}^t\boldsymbol{w}^\star) + \frac{a(t)}{m}\sum_{j=1}^m \frac{1}{n}\sigma(\boldsymbol{X}^t\boldsymbol{w}_i^t)^\top\sigma(\boldsymbol{X}^t\boldsymbol{w}_j^t)$$

$$\xrightarrow{n,d\to\infty} -\mathbb{E}[\sigma(\widehat{Z}^{\boldsymbol{X}^t\boldsymbol{w}_i^t})\sigma_\star(\widehat{Z}^{\boldsymbol{X}^t\boldsymbol{w}_\star})] + a(t)\frac{m-1}{m}\mathbb{E}[\sigma(\widehat{Z}^{\boldsymbol{X}^t\boldsymbol{w}_i^t})\sigma(\widehat{Z}^{\boldsymbol{X}^t\boldsymbol{w}_j^t})] + \frac{a(t)}{m}\mathbb{E}[\sigma(\widehat{Z}^{\boldsymbol{X}^t\boldsymbol{w}_i^t})^2]$$

$$= -I_R(t; \sigma, \sigma_\star) + a(t)I_Q(t; \sigma, \sigma) - m^{-1}a(t)I_Q(t; \sigma, \sigma) + m^{-1}a(t)s_4.$$

Since this form is independent of $i$, we can write

$$a(t+1) = a(t) + \gamma m^{-1}\big\{I_R(t; \sigma, \sigma_\star) - a(t)I_Q(t; \sigma, \sigma)\big\} + \gamma m^{-2}a(t)\big\{I_Q(t; \sigma, \sigma) - s_4\big\},$$

where we can see this matches the model (8) by Hermite expansion. This completes the proof. $\square$

### C.3. Proof of Proposition 1

This limiting ODE is directly derived from the standard Euler method. Here, by using Lemma 9 (presented in Section C.4) and the relation $R_\tau^2 = Q_\tau$, we can omit the variable $Q_\tau$. Then, we can formulate it as the following lemma.

**Lemma 8.** *Let $R_\tau^m := R^m(\lfloor m\tau/\gamma \rfloor), a_\tau^m := a^m(\lfloor m\tau/\gamma \rfloor)$. Then, for any finite $\tau \geq 0$, asymptotic equalities*

$$\lim_{m\to\infty} R_\tau^m = R_\tau, \quad \lim_{m\to\infty} a_\tau^m = a_\tau$$

*hold for $R_\tau, a_\tau$ satisfying the ODE (6).*

*Proof of Lemma 8.* Since $f$ and $g$ in (6) are analytic functions, the result immediately holds from the standard discussion of numerical analysis, e.g., Theorem 2.4 in Atkinson et al. (2009). $\square$

Finally, by combining the results, we can prove Proposition 1:

*Proof of Proposition 1.* It immediately holds by Proposition 6 and Lemma 8. $\square$

### C.4. Reduction of $Q_\tau$

We provide the following lemma, which allows us to omit the variable $Q_\tau$.

**Lemma 9.** *For any $\tau \geq 0$, it holds that*

$$Q_\tau = R_\tau^2.$$

*Proof.* Let $U_\tau = Q_\tau - R_\tau^2$. Then, it follows that

$$
\begin{aligned}
\frac{dU_\tau}{d\tau} &= \frac{dQ_\tau}{d\tau} - 2R_\tau \frac{dR_\tau}{d\tau} \\
&= 2a_\tau R_\tau \left(R_\tau^2 - Q_\tau\right) \sum_{k=0}^{\infty} (k+1) \cdot (k+1)! c_{\star,k+1} c_{k+1} R_\tau^k \\
&\quad - 2a_\tau^2 \left(R_\tau^2 - Q_\tau\right)\left(1 - Q_\tau\right) \sum_{k=0}^{\infty} (k+1) \cdot (k+1)! c_{k+1}^2 Q_\tau^k \\
&= 2a_\tau \left(R_\tau^2 - Q_\tau\right)\Big\{ R_\tau \sum_{k=0}^{\infty} (k+1) \cdot (k+1)! c_{\star,k+1} c_{k+1} R_\tau^k \\
&\quad + a_\tau \left(1 - Q_\tau\right) \sum_{k=0}^{\infty} (k+1) \cdot (k+1)! c_{k+1}^2 Q_\tau^k \Big\} \\
&= -2 U_\tau V_\tau,
\end{aligned}
$$

where

$$V_\tau = a_\tau \Big\{ R_\tau \sum_{k=0}^{\infty} (k+1) \cdot (k+1)! c_{\star,k+1} c_{k+1} R_\tau^k + a_\tau \left(1 - Q_\tau\right) \sum_{k=0}^{\infty} (k+1) \cdot (k+1)! c_{k+1}^2 Q_\tau^k \Big\}.$$

Then we obtain

$$U_\tau = U_0 \exp\left( -2 \int_0^\tau V_s ds \right) = 0,$$

since $U_0 = 0$. $\square$

## D. Derivation of ODE via population gradient

We show that we can derive the same ODE as (6) by leveraging the gradient flow of the population loss. In particular, we consider an expected loss with the two-layer neural network $f(\boldsymbol{x}; \boldsymbol{a}, \boldsymbol{W}) = \frac{1}{m} \sum_{i=1}^{m} a_i \sigma(\langle \boldsymbol{w}_i, \boldsymbol{x} \rangle / \sqrt{d})$, and define a solution of an ODE defined by the gradient of the expected loss.

For time $\tau > 0$, we define a solution $(\check{\boldsymbol{a}}_\tau, \check{\boldsymbol{W}}_\tau)_{\tau \geq 0}$ with initialization

$$\check{a}_{i,0} = \bar{a}, \quad \check{\boldsymbol{w}}_{i,0}, \check{\boldsymbol{w}}_\star \overset{\text{i.i.d.}}{\sim} \text{Unif}\left(\mathbb{S}^{d-1}(\sqrt{d})\right),$$

and the gradient of the population loss:

$$\mathcal{L}(\check{\boldsymbol{a}}_\tau, \check{\boldsymbol{W}}_\tau) = \frac{1}{2}\mathbb{E}\Big[\big(\sigma(\langle \check{\boldsymbol{w}}_\star, \boldsymbol{x} \rangle / \sqrt{d}) - \frac{1}{m}\sum_{i=1}^{m}\check{a}_{i,\tau}\sigma(\langle \check{\boldsymbol{w}}_{i,\tau}, \boldsymbol{x} \rangle / \sqrt{d})\big)^2\Big]$$

$$= \frac{1}{2}\Big(\frac{1}{m^2}\sum_{i,j=1}^{m}\check{a}_{i,\tau}\check{a}_{j,\tau}Y(\langle \check{\boldsymbol{w}}_{i,\tau}, \check{\boldsymbol{w}}_{j,\tau} \rangle / d) - \frac{2}{m}\sum_{i}\check{a}_{i,\tau}S(\langle \check{\boldsymbol{w}}_\star, \boldsymbol{w}_{i,\tau} \rangle / d)\Big) + \text{const},$$

with

$$S(z) = \sum_{k=1}^{\infty} c_{\star,k} c_k z^k, \quad Y(z) = \sum_{k=1}^{\infty} c_k^2 z^k.$$

We then calculate the gradients as follows:

$$\frac{d\check{a}_{i,\tau}}{d\tau} = -m\partial_{\check{a}_{i,\tau}}\mathcal{L}(\check{\boldsymbol{a}}_\tau, \check{\boldsymbol{W}}) = -\frac{1}{m}\sum_{j=1}^{m}\check{a}_{j,\tau}Y(\langle \check{\boldsymbol{w}}_i, \check{\boldsymbol{w}}_j \rangle / d) + S(\langle \check{\boldsymbol{w}}_\star, \check{\boldsymbol{w}}_i \rangle / d),$$

$$\frac{d\check{\boldsymbol{w}}_{i,\tau}}{d\tau} = -md\Big(\boldsymbol{I}_d - \frac{\check{\boldsymbol{w}}_{i,\tau}\check{\boldsymbol{w}}_{i,\tau}^\top}{d}\Big)\nabla_{\check{\boldsymbol{w}}_{i,\tau}}\mathcal{L}(\check{\boldsymbol{a}}_\tau, \check{\boldsymbol{W}}_i)$$

$$= -\frac{\check{a}_{i,\tau}}{m}\sum_{j=1}^{k}\check{a}_{j,\tau}Y'(\langle \check{\boldsymbol{w}}_{i,\tau}, \check{\boldsymbol{w}}_{j,\tau} \rangle / d) \cdot \big(\check{\boldsymbol{w}}_{j,\tau} - \langle \check{\boldsymbol{w}}_{i,\tau}, \check{\boldsymbol{w}}_{j,\tau} \rangle \check{\boldsymbol{w}}_{i,\tau}\big)$$

$$+ \check{a}_{i,\tau}S'(\langle \boldsymbol{w}_\star, \check{\boldsymbol{w}}_{i,\tau} \rangle / d) \cdot \big(\boldsymbol{w}_\star - \langle \check{\boldsymbol{w}}_\star, \check{\boldsymbol{w}}_{i,\tau} \rangle \check{\boldsymbol{w}}_{i,\tau}\big).$$

To simplify the form, we define the alignments $\check{R}_{i,\tau}, \check{Q}_{ij,\tau}$ as

$$\check{R}_{i,\tau} = \frac{\langle \check{\boldsymbol{w}}_\star, \check{\boldsymbol{w}}_{i,\tau} \rangle}{d}, \quad \check{Q}_{ij,\tau} = \frac{\langle \check{\boldsymbol{w}}_{i,\tau}, \check{\boldsymbol{w}}_{j,\tau} \rangle}{d}.$$

We can derive ODEs for these order parameters.

$$\frac{d\check{a}_{i,\tau}}{d\tau} = -\frac{1}{m}\sum_{j=1}^{m}\check{a}_{j,\tau}Y(\check{Q}_{ij,\tau}) + S(\check{R}_{i,\tau}),$$

$$\frac{d\check{R}_{i,\tau}}{d\tau} = -\frac{\check{a}_{i,\tau}}{m}\sum_{j=1}^{m}\check{a}_{j,\tau}Y'(\check{Q}_{ij,\tau})(\check{R}_{j,\tau} - \check{Q}_{ij,\tau}\check{R}_{i,\tau}) + \check{a}_{i,\tau}S'(\check{R}_{i,\tau})(1 - \check{R}_{i,\tau}^2),$$

$$\frac{d\check{Q}_{ij,\tau}}{d\tau} = -\frac{\check{a}_{i,\tau}}{m}\sum_{k=1}^{m}\check{a}_{k,\tau}Y'(\check{Q}_{ik,\tau})(\check{Q}_{jk,\tau} - \check{Q}_{ik,\tau}\check{Q}_{ij,\tau}) - \frac{\check{a}_{j,\tau}}{m}\sum_{k=1}^{m}\check{a}_{k,\tau}Y'(\check{Q}_{jk,\tau})(\check{Q}_{ik,\tau} - \check{Q}_{jk,\tau}\check{Q}_{ij,\tau})$$

$$+ \check{a}_{i,\tau}S'(\check{R}_{i,\tau})(\check{R}_{j,\tau} - \check{R}_{i,\tau}\check{Q}_{ij,\tau}) + \check{a}_{j,\tau}S'(\check{R}_{j,\tau})(\check{R}_{i,\tau} - \check{R}_{j,\tau}\check{Q}_{ij,\tau}).$$

(16)

With the symmetric initialization, the following holds:

$$\check{a}_{i,\tau} = \check{a}_\tau, \quad \check{R}_{i,\tau} = \check{R}_\tau, \quad \check{Q}_{ij,\tau} = \check{Q}_\tau (i \neq j).$$

Then, the ODE (16) can be further simplified as follows when $m \to \infty$:

$$
\frac{d\check{a}_\tau}{d\tau} = S(\check{R}_\tau) - \check{a}_\tau Y(\check{Q}_\tau),
$$

$$
\frac{d\check{R}_\tau}{d\tau} = \check{a}_\tau(1 - \check{R}_\tau^2)S'(\check{R}_\tau) - \check{a}_\tau^2(1 - \check{Q}_\tau)\check{R}_\tau Y'(\check{Q}_\tau),
$$

$$
\frac{d\check{Q}_\tau}{d\tau} = 2\big(\check{a}_\tau(1 - \check{Q}_\tau)\check{R}_\tau S'(\check{R}_\tau) - \check{a}_\tau^2(1 - \check{Q}_\tau)\check{Q}_\tau Y'(\check{Q}_\tau)\big).
$$

Since we have

$$
\begin{aligned}
\frac{d}{d\tau}(\check{Q}_\tau - \check{R}_\tau^2) &= \frac{d\check{Q}_\tau}{d\tau} - 2\check{R}_\tau\frac{d\check{R}_\tau}{d\tau} \\
&= 2\big(\check{a}_\tau(1 - \check{Q}_\tau)\check{R}_\tau S'(\check{R}_\tau) - \check{a}_\tau^2(1 - \check{Q}_\tau)\check{Q}_\tau Y'(\check{Q}_\tau)\big) \\
&\quad - 2\check{R}_\tau\big(\check{a}_\tau(1 - \check{R}_\tau^2)S'(\check{R}_\tau) - \check{a}_\tau^2(1 - \check{Q}_\tau)\check{R}_\tau Y'(\check{Q}_\tau)\big) \\
&= -2\check{a}_\tau\check{R}_\tau S'(\check{R}_\tau)(\check{Q}_\tau - \check{R}_\tau^2) - 2\check{a}_\tau^2(1 - \check{Q}_\tau)Y'(\check{Q}_\tau)(\check{Q}_\tau - \check{R}_\tau^2),
\end{aligned}
$$

we obtain $\check{Q}_\tau = \check{R}_\tau^2$. With defining $T(\check{R}_\tau) = U(\check{R}_\tau^2) = \sum_{k=1}^\infty c_k^2 \check{R}_\tau^{2k}$, from $\check{R}_\tau U'(\check{Q}_\tau) = \frac{1}{2}T'(\check{R}_\tau)$, we obtain

$$
\frac{d\check{R}_\tau}{d\tau} = \check{a}_\tau(1 - \check{R}_\tau^2)S'(\check{R}_\tau) - \frac{1}{2}\check{a}_\tau^2(1 - \check{R}_\tau^2)T'(\check{R}_\tau),
$$

$$
\frac{d\check{a}_\tau}{d\tau} = S(\check{R}_\tau) - \check{a}_\tau T(\check{R}_\tau).
$$

This exactly matches the ODE (6).

# E. Origin of the fast-slow dynamics

The fast-slow structure observed in the dynamics of (6) is primarily motivated by numerical experiments, but it can be partially justified theoretically in specific regimes. In particular, the flow initially evolves purely in the $R$-direction at $R = 0$, since the $a$-component of the vector field vanishes there. Moreover, when $|R|$ is small, and the trajectory evolves near the nontrivial branch of the critical manifold, the Jacobian exhibits a strong separation of eigenvalues, with the fast eigendirection nearly aligned with the $R$-axis. In this regime, the fast-slow ansatz adopted in the main text is therefore theoretically justified, which is especially relevant for the feature unlearning scenarios studied in this work.

### E.1. Initial transient and quasi-frozen $a_\tau$.

We consider the two-dimensional ODE (6). Then, from $S(0) = 0, T(0) = 0$, it holds that

$$
g(0, \bar{a}) = S(0) - \bar{a}T(0) = 0.
$$

Moreover, since $S'(0) = 2c_{\star,1}c_1$ and $T'(0) = 0$, we obtain

$$
f(0, \bar{a}) = \frac{1}{2}\bar{a}\big(2S'(0) - \bar{a}T'(0)\big) = \bar{a}c_{\star,1}c_1 > 0.
$$

Therefore, at the initial time $\tau = 0$, one has $\dot{a}_\tau = 0$ while $\dot{R}_\tau > 0$ holds.

Expanding the vector field for small $f(R, a)$ (with $a$ treated as $O(1)$ during this short transient), we obtain

$$
f(R, a) = f(0, a) + O(R), \quad g(R, a) = g_R(0, a)R + O(R^2),
$$

and hence it holds that

$$
\frac{da_\tau}{dR_\tau} = \frac{g(R_\tau, a_\tau)}{f(R_\tau, a_\tau)} = O(R_\tau).
$$

This shows that $a_\tau$ remains approximately frozen while $R_\tau$ moves rapidly away from 0, resulting in a fast relaxation toward $\mathcal{S}$.

### E.2. Critical manifold and scale separation for small $|R|$

In this section, we discuss that the fast dynamics primarily drive the development of $R_\tau$, and this development is directed toward the critical manifold $\mathcal{S}$. The critical manifold (or $R$-nullcline) is defined by $f(R,a) = 0$ and consists of three branches: $a = 0$, $R = \pm 1$, and the nontrivial branch $a = h(R)$ defined in Assumption 7. We focus on the last one, which is relevant to the feature unlearning phenomenon observed in numerical experiments.

Recall that $k_0 \geq 2$ denotes the smallest integer such that $c_{\star,k_0} c_{k_0} \neq 0$, and $k_1 \geq 2$ denotes the smallest integer such that $c_{k_1}^2 \neq 0$. Although the correlation functions satisfy $S(R) = O(R)$ and $T(R) = O(R^2)$ as $R \to 0$, their leading-order derivatives are governed by these minimal indices. In particular, as $R \to 0$, we obtain

$$S'(R) = \Theta(1), \qquad S''(R) = \Theta(R^{k_0-2}), \qquad T'(R) = \Theta(R), \qquad T''(R) = \Theta(1). \tag{17}$$

Along the nontrivial critical manifold $a = h(R) = 2S'(R)/T'(R)$, the estimates (17) immediately imply

$$a = h(R) = \Theta(R^{-1}), \qquad (R \to 0), \tag{18}$$

so that the amplitude $a$ diverges algebraically as $R \to 0$.

To quantify the resulting time-scale separation, we consider the Jacobian of the ODE (6) at $(R_\tau, a_\tau) = (R, h(R))$,

$$J(R,a) = \begin{pmatrix} f_R & f_a \\ g_R & g_a \end{pmatrix}, \quad g_a = -T(R), \quad g_R = S'(R) - aT'(R), \quad f_a = (1 - R^2)g_R,$$

and the remaining entry is given by:

$$f_R = \frac{1}{2}a\Big[(-2R)\big(2S'(R) - aT'(R)\big) + (1 - R^2)\big(2S''(R) - aT''(R)\big)\Big]. \tag{19}$$

On the critical manifold $a = h(R)$, the first term in (19) vanishes identically, yielding

$$f_R = \frac{1}{2}a(1 - R^2)\big(2S''(R) - aT''(R)\big).$$

Using (17) and (18), the dominant contribution arises from the $-a^2 T''(R)$ term, so that we have

$$f_R = \Theta(a^2) = \Theta(R^{-2}), \qquad (R \to 0). \tag{20}$$

In contrast, the remaining Jacobian entries scale as

$$g_a = -T(R) = \Theta(R^2), \qquad g_R = S'(R) - aT'(R) = \Theta(1), \qquad f_a = (1 - R^2)g_R = \Theta(1). \tag{21}$$

Therefore, along the nontrivial critical manifold, the Jacobian has the schematic structure

$$J(R, h(R)) = \begin{pmatrix} \Theta(R^{-2}) & \Theta(1) \\ \Theta(1) & \Theta(R^2) \end{pmatrix}, \qquad (R \to 0).$$

Treating the large entry $f_R$ as dominant, the eigenvalues satisfy

$$\lambda_f = f_R + O(1) = \Theta(R^{-2}), \qquad \lambda_s = g_a - \frac{f_a g_R}{f_R} + O(R^2) = O(R^2). \tag{22}$$

Thus, the local time-scale ratio obeys

$$\frac{|\lambda_s|}{|\lambda_f|} = O(R^4) \ll 1, \qquad (R \to 0).$$

This establishes a pronounced scale separation of two eigenvalues along the critical manifold for sufficiently small $|R|$, especially when feature unlearning occurs.

We also discuss the fast eigenvector alignment with the $R$-direction. Let $\boldsymbol{v}_f = (v_{f,R}, v_{f,a})^\top$ denote the eigenvector associated with $\lambda_f$. Writing the eigenvector equation

$$(f_R - \lambda_f)v_{f,R} + f_a v_{f,a} = 0, \qquad g_R v_{f,R} + (g_a - \lambda_f)v_{f,a} = 0,$$

and using $\lambda_f \approx f_R$ gives the following from the second equation:

$$\frac{v_{f,a}}{v_{f,R}} = -\frac{g_R}{g_a - \lambda_f} = \Theta\left(\frac{1}{|f_R|}\right) = \Theta(R^2), \qquad (R \to 0),$$

since $g_R = \Theta(1)$ and $g_a - \lambda_f = \Theta(R^{-2})$ by (20), (21), and (22). Thus, the fast eigendirection satisfies that $v_f$ is almost parallel to $e_R$ up to $\Theta(R^2)$, justifying the interpretation that $R$ is the fast variable near the critical manifold for small $|R|$.

## F. Proof for Section 5

### F.1. Proof of Theorem 2 and Theorem 3

The proof of the main theorems is based on Lobry et al. (1998). It proceeds as follows:

(1) verify (H1) - (H5) of Lobry et al. (1998) hold for the model (7);

(2) apply Theorem 1 of Lobry et al. (1998) to the system.

In preparation, we introduce several notations related to the critical manifold $\mathcal{S}$. Specifically, $\mathcal{S}$ can be decomposed as $\mathcal{S} = \mathcal{S}_0^+ \sqcup \mathcal{S}_0^- \sqcup \mathcal{S}_1$, where $\mathcal{S}_1 := \{(R, a) \in \{-1, 1\} \times \mathbb{R}\}$ and $\mathcal{S}_0 := \mathcal{S}_0^+ \sqcup \mathcal{S}_0^-$ with

$$\mathcal{S}_0^+ := \{(R, a) \in (0, 1) \times \mathbb{R} \mid 2S'(R) - aT'(R) = 0\},$$
$$\mathcal{S}_0^- := \{(R, a) \in (-1, 0) \times \mathbb{R} \mid 2S'(R) - aT'(R) = 0\}.$$

Since $\bar{f}(0, \bar{a}) = \bar{a}c_{\star,1}c_1/\Lambda_f > 0$ from Assumptions 4 and 5, we expect $R_{\tau_s}^\varepsilon$ rapidly increases and rides onto $\mathcal{S}_0^+$ in the fast flow.

First, we show the following technical lemma.

**Lemma 10.** *For $h, \alpha : (0, 1) \to \mathbb{R}$ defined in Assumption 7, the following holds.*

$$h(R) = \frac{c_{\star,1}}{c_1} R^{-1} + o(R^{-1}),$$
$$\alpha(R) = -2(k_0 - 1)k_0! c_{\star,k_0} c_{k_0} c_1^2 R^{k_0+1} + 2(k_1 - 1)k_1! c_{\star,1} c_1 c_{k_1}^2 R^{2k_1} + O(R^{\max\{k_0+1, 2k_1\}+1})$$

*as $R \to +0$.*

*Proof.* These follow from direct expansion:

$$h(R) = \frac{2S'(R)}{T'(R)}$$
$$= \frac{2c_{\star,1}c_1 + O(R)}{2c_1^2 R + O(R^2)}$$
$$= \frac{c_{\star,1}}{c_1} R^{-1} + o(R^{-1}), \quad R \to +0,$$

and

$$\alpha(R) = S(R)T'(R) - 2S'(R)T(R)$$
$$= (c_{\star,1}c_1 R + k_0! c_{\star,k_0} c_{k_0} R^{k_0} + O(R^{k_0+1}))(2c_1^2 R + 2k_1 \cdot k_1! c_{k_1}^2 R^{2k_1-1} + O(R^{2k_1}))$$
$$\quad - 2(c_{\star,1}c_1 + k_0 \cdot k_0! c_{\star,k_0} c_{k_0} R^{k_0-1} + O(R^{k_0}))(c_1^2 R^2 + k_1! c_{k_1}^2 R^{2k_1} + O(R^{2k_1+1}))$$
$$= -2(k_0 - 1)k_0! c_{\star,k_0} c_{k_0} c_1^2 R^{k_0+1} + 2(k_1 - 1)k_1! c_{\star,1} c_1 c_{k_1}^2 R^{2k_1} + O(R^{\max\{k_0+1, 2k_1\}+1}),$$

as $R \to +0$. $\qquad\square$

Now we state the following lemma, which states that Theorem 1 of Lobry et al. (1998) can be applied to the system (7). In the following, we introduce $(\widehat{R}_{\tau_f})_{\tau_f \geq 0}$ and $(\widehat{a}_{\tau_s})_{\tau_s \geq 0}$ as a solution of a differential equation, then study its dynamics.

**Lemma 11.** *Under Assumptions 3-7, for an open set $D = (-1, 1) \times (0, \infty) \subset \mathbb{R}^2$ and any $M > \bar{a}$, all the following hypotheses (H1) - (H5) hold.*

*(H1) For any fixed $a \in (0, \infty)$, the fast equation*

$$\frac{d\widehat{R}_{\tau_f}}{d\tau_f} = \bar{f}(\widehat{R}_{\tau_f}, a) \quad \tau_f = \tau_s/\varepsilon \tag{23}$$

*has a unique solution $(\widehat{R}_{\tau_f})_{\tau_f}$ with prescribed initial conditions.*

*(H2) There exits some $\delta > 0$ such that, for $I_a = [\bar{a} - \delta, M]$, there exits some function $\xi : I_a \to \mathbb{R}$ such that for any $a \in I_a$, $R = \xi(a)$ is an isolated root of an equation $\bar{f}(R, a) = 0$ and $\mathcal{L} := \{(\xi(a), a); a \in I_a\} \subset D$ holds.*

*(H3) For any $a \in I_a$, $R = \xi(a)$ is an asymptotically stable equilibrium point of the fast equation, and we can take the basin of attraction of $R = \xi(a)$ uniformly over $I_a$.*

*(H4) The slow equation*

$$\frac{d\widehat{a}_{\tau_s}}{d\tau_s} = \bar{g}(\xi(\widehat{a}_{\tau_s}), \widehat{a}_{\tau_s}) \tag{24}$$

*defined on $\mathring{I}_a = \operatorname{int} I_a$ has a unique solution $(\widehat{a}_{\tau_s})_{\tau_s}$ with prescribed initial conditions.*

*(H5) $\bar{a} \in \mathring{I}_a$ holds, and the point $R(0) = 0$ is in the basin of attraction of the equilibrium point $R = \xi(\bar{a})$ in the fast equation (23).*

*Proof.* **(H1)** For fixed $a$, $\bar{f}(R, a)$ is a polynomial of $R$, and especially, $C^\infty$ in $D$. So, (H1) follows from the Picard-Lindelöf theorem.

**(H2)** From Assumption 7 and Lemma 10, if we take $\delta > 0$ sufficiently small, $I_a := [\bar{a} - \delta, M] \subset h((0, R^\star))$ holds. Since $1 - R^2 > 0$ and $T'(R) \neq 0$ when $R \neq 0$, we have

$$\bar{f}(R, a) = 0 \iff 2S'(R) - T'(R)a = 0 \iff a = h(R) = \frac{2S'(R)}{T'(R)}.$$

From $I_a \subset h((0, R^\star))$ and Lemma 10, there exists some closed interval $I_R \subset (0, R^\star)$ such that $h : I_R \to I_a$ is monotonically decreasing, and therefore, bijective. Thus, the inverse function $\xi := h^{-1} : I_a \to I_R$ exists. Also, by its definition, $\bar{f}(\xi(a), a) = 0$ holds for $a \in I_a$, and we obtain $\mathcal{L} = \{(R, a); \bar{f}(\xi(a), a) = 0, a \in I_a\} \subset D$ from $I_R \subset (0, R^\star) \subset (-1, 1)$ and $I_a = [\bar{a} - \delta, M) \subset (0, \infty)$.

**(H3)** We obtain that, for $R = \xi(a), a \in I_a$

$$\begin{aligned}
\partial_R \bar{f}(R, a)\big|_{R=\xi(a)} &= \frac{1}{2\Lambda_f} a(1 - R^2)\{2S''(R) - aT''(R)\} \\
&= \frac{a(1 - R^2)}{\Lambda_f} \cdot \frac{S''(R)T'(R) - S'(R)T''(R)}{T'(R)} \\
&= \frac{1}{\Lambda_f} S'(R)(1 - R^2)h'(R) \\
&= \frac{1}{2\Lambda_f}(1 - R^2)T'(R)h(R)h'(R) \\
&= \frac{1}{2\Lambda_f}(1 - \xi(a)^2)T'(\xi(a))h(\xi(a))h'(\xi(a)).
\end{aligned}$$

From $I_a \subset h((0, R^\star))$, we have $\xi(a) \in (0, R^\star)$ for any $a \in I_a$. Then, it follows that

$$1 - \xi(a)^2 > 0, \quad T'(\xi(a)) > 0, \quad h(\xi(a)) > 0, \quad h'(\xi(a)) < 0.$$

Therefore, $\partial_R \bar{f}(R,a)\big|_{R=\xi(a)} < 0$ holds for any $a \in I_a$. Hence, for any fixed $a \in I_a$, $R = \xi(a)$ is an asymptotically stable equilibrium point of the fast equation. Now, from the compactness of $I_a$, there exists some $\theta > 0$, such that, for any $a \in I_a$ and $R \in (\xi(a) - \theta, \xi(a) + \theta)$, $\partial_R \bar{f}(R,a) < 0$ holds. This means that we can take the basin of attraction of $R = \xi(a)$ uniformly over $I_a$.

**(H4)** We can prove $\xi : \mathring{I}_a \to \mathring{I}_R$ is $C^\infty$ from the inverse function theorem. Then, the function $a \mapsto \bar{g}(\xi(a), a)$ is also $C^\infty$ in $\mathring{I}_a$. Hence, the slow equation (24) has a unique solution from the Picard-Lindelöf theorem.

**(H5)** By the definition of $I_a$, $\bar{a} \in \mathring{I}_a$. Also, for any $a \in \mathring{I}_a$, $\bar{f}(R,a) > 0$ for $0 \le R < \xi(a)$, and $\bar{f}(\xi(a), a) = 0$. This means $R = 0$ is in the basin of attraction of $R = \xi(a)$. $\qquad\square$

Together with Lemma 11 and Theorem 1 of Lobry et al. (1998), we obtain the following theorem:

**Theorem 12.** *Let each solution of the fast equation* (23) *with an initial value* 0, *and the slow equation* (24) *with an initial value* $\bar{a}$, *be* $\widehat{R}_{\tau_f}$, $\widehat{a}_{\tau_s}$, *respectively. Let* $T > 0$ *be a maximal positive interval of definition of the slow equation* (24). *Then, under Assumptions 3-7, the following holds; for any* $\eta > 0$, *there exists some* $\epsilon_\eta > 0$ *with the property that, if* $\epsilon < \epsilon_\eta$, *the solution of* (7) *is defined for at least* $\tau_s \in [0, T]$, *and there exits* $L > 0$ *such that* $\varepsilon L < \eta$, $|R^\varepsilon_{\varepsilon\tau_f} - \widehat{R}_{\tau_f}| < \eta$ *for* $0 \le \tau_f \le L$, $|R^\varepsilon_{\tau_s} - \xi(\widehat{a}_{\tau_s})| < \eta$ *for* $\varepsilon L \le \tau_s \le T$ *and* $|a^\varepsilon_{\tau_s} - \widehat{a}_{\tau_s}| < \eta$ *for* $0 \le \tau_s \le T$.

Next, based on the result above, we study the asymptotic behavior of $(\xi(\widehat{a}_{\tau_s}), \widehat{a}_{\tau_s})$ for $\tau_s \to \infty$ when the slow equation (24) is defined on $[\bar{a} - \delta, \infty)$ with $\delta > 0$ used in (H2) of Lemma 11. We show the following lemma.

**Lemma 13.** *For the solution* $\widehat{a}_{\tau_s}$ *of the slow equation* (24) *defined on* $[\bar{a} - \delta, \infty)$ *with* $\delta > 0$ *used in (H2) of Lemma 11, the following holds:*

$$\lim_{\tau_s \to \infty} \widehat{a}_{\tau_s} = \infty, \quad \lim_{\tau_s \to \infty} \xi(\widehat{a}_{\tau_s}) = 0, \quad \lim_{\tau_s \to \infty} \widehat{a}_{\tau_s} \xi(\widehat{a}_{\tau_s}) = \frac{c_{\star,1}}{c_1}, \tag{25}$$

*where the scaling law changes for each case of Assumption 6:*

(i) $\widehat{a}_{\tau_s} = \Theta(\tau_s^{1/2k_1}), \quad \xi(\widehat{a}_{\tau_s}) = \Theta(\tau_s^{-1/2k_1}),$

(ii) $\widehat{a}_{\tau_s} = \Theta(\tau_s^{1/(k_0+1)}), \quad \xi(\widehat{a}_{\tau_s}) = \Theta(\tau_s^{-1/(k_0+1)}),$

*when* $\tau_s \to \infty$.

*Proof.* For $h : h^{-1}([\bar{a} - \delta, \infty)) \to [\bar{a} - \delta, \infty)$, from Lemma 10, we have $R \to +0 \iff h(R) \to \infty$. Thus, the inverse function $R = h^{-1}(a) = \xi(a)$ is

$$\xi(a) = \frac{c_{\star,1}}{c_1} a^{-1} + o(a^{-1}), \quad a \to \infty. \tag{26}$$

We now consider the slow equation

$$\frac{d\widehat{a}_{\tau_s}}{d\tau_s} = \bar{g}(\xi(\widehat{a}_{\tau_s}), \widehat{a}_{\tau_s}) = \frac{\alpha(\xi(\widehat{a}_{\tau_s}))}{\Lambda_s T'(\xi(\widehat{a}_{\tau_s}))}, \quad \widehat{a}_0 = \bar{a} > 0. \tag{27}$$

Combining Lemma 10, (26) and $T'(R) = 2c_1^2 R + O(R^2)$, in the case (i) of Assumption 6, we have

$$\begin{aligned}
\frac{d\widehat{a}_{\tau_s}}{d\tau_s} &= \frac{1}{\Lambda_s} \cdot (k_1-1)k_1! c_{k_1}^2 \frac{c_{\star,1}}{c_1} (c_{\star,1}/c_1)^{2k_1} \widehat{a}_{\tau_s}^{-2k_1+1} + o(\widehat{a}_{\tau_s}^{-2k_1+1}) \\
&= K \widehat{a}_{\tau_s}^{-2k_1+1} + o(\widehat{a}_{\tau_s}^{-2k_1+1}), \quad \widehat{a}_{\tau_s} \to \infty,
\end{aligned}$$

for some constant $K > 0$. From (27) and Assumption 7, $\widehat{a}(\tau_s)$ monotonically increases along $\mathcal{S}$. Then, $\widehat{a}_{\tau_s}$ is defined for $\tau_s \ge 0$, since $\widehat{a}_{\tau_s} \in [\bar{a} - \delta, \infty)$ holds for $\tau_s \ge 0$. By standard comparison arguments for scalar ODEs, we have $\widehat{a}_{\tau_s} \to \infty$ with $\widehat{a}_{\tau_s} = \Theta(\tau_s^{1/(2k_1)})$ when $\tau_s \to \infty$. From (26), $\xi(\widehat{a}_{\tau_s}) \to 0$ with $\xi(\widehat{a}_{\tau_s}) = \Theta(\tau_s^{-1/2k_1})$ also holds. The last equality of (25) follows from $a\xi(a) = c_{\star,1}/c_1 + o(1)$, $a \to \infty$. We can derive another scaling law for (ii) in a similar way. $\qquad\square$

Since we can take $T > 0$ arbitrarily large if we set $M > 0$ sufficiently large, we obtain the following statement.

**Corollary 14.** $R^0_{\tau_s}, a^0_{\tau_s}$ *can be defined for any* $\tau_s \in [0, \infty)$*, and it holds that*

$$a^0_{\tau_s} = \widehat{a}_{\tau_s}, \quad R^0_{\tau_s} = \xi(\widehat{a}_{\tau_s}),$$

*for any* $\tau_s > 0$.

*Proof.* Fix $\tau_s > 0$. With sufficiently large $M$, we can set $T > \tau_s$. Then, from Theorem 12, for any $0 < \eta < \tau_s$, with sufficiently small $\varepsilon > 0$, $|R^\varepsilon_{\tau_s} - \xi(\widehat{a}_{\tau_s})| < \eta$ holds. This directly implies $\lim_{\varepsilon \to +0} |R^\varepsilon_{\tau_s} - \xi(\widehat{a}_{\tau_s})| = 0$. We can prove similarly in the case of $a^\varepsilon_{\tau_s}$. $\qquad\square$

Together with these results, we finally prove the main theorems.

*Proof of Theorem 2.* From Lemma 13 and Corollary 14, we obtain

$$\lim_{\tau_s \to \infty} a^0_{\tau_s} = \lim_{\tau_s \to \infty} \widehat{a}_{\tau_s} = \infty, \quad \lim_{\tau_s \to \infty} R^0_{\tau_s} = \lim_{\tau_s \to \infty} \xi(\widehat{a}_{\tau_s}) = 0,$$

and

$$\lim_{\tau_s \to \infty} R^0_{\tau_s} a^0_{\tau_s} = \lim_{\tau_s \to \infty} \xi(\widehat{a}_{\tau_s}) \widehat{a}_{\tau_s} = \frac{c_{\star,1}}{c_1}.$$

$\qquad\square$

Quite similarly, Theorem 3 directly follows from the combination of Lemma 13 and Corollary 14.

### F.2. Proof of Theorem 4

*Proof.* For $R < R^{\star\star}$, we have

$$H(R) - h(R) = \frac{ST' - 2S'T}{TT'} = \frac{\alpha}{TT'} > 0,$$

and thus $h(R) < H(R)$.

For $(R, a) \in K$, we have $g(R, a) = S(R) - aT(R) = T(R)(H(R) - a) \geq 0$.

We claim that the trajectory does not escape $K$ once inside. This can be shown by checking that the vector field does not point out of $K$ at the boundary of $K$, as illustrated schematically in Figure 5.

- When $a = \bar{a}$, $g(R, \bar{a}) > 0$ as shown above.

- When $R = 0$, $f(0, a) = aS'(0) > 0$.

- When $a = H(R)$, $g(R, a) = 0$ and

$$f(R, a) = \frac{1}{2}a(1 - R^2)T'(R)(h(R) - a) = \frac{1}{2}a(1 - R^2)T'(R)(h(R) - H(R)) < 0.$$

- When $R = R^{\star\star}$ and $a \geq \bar{a}$,

$$f(R^{\star\star}, a) = \frac{1}{2}a(1 - (R^{\star\star})^2)T'(R)(h(R^{\star\star}) - a) \leq \frac{1}{2}a(1 - (R^{\star\star})^2)T'(R)(h(R^{\star\star}) - \bar{a}) < 0.$$

Since the initial value $(0, \bar{a}) \in K$, the trajectory remains in $K$ for all time. Since there is no fixed point either inside $K$ or on its boundary and $da_\tau/d\tau = g(R_\tau, a_\tau) \geq 0$, we have $a_\tau \to \infty$ and thus $R_\tau \to 0$ as $\tau \to \infty$. $\qquad\square$

### F.3. When feature unlearning does not occur

We complement Theorem 2 by describing when the fast–slow dynamics leads to feature learning rather than feature unlearning. Throughout this subsection, we focus on the positive branch $\mathcal{S}_0^+$, since under Assumptions 4 and 5 the initial fast flow moves from $R = 0$ to the positive-$R$ direction. Recall that, on $\mathcal{S}_0^+$, the critical manifold is parameterized by

$$a = h(R) = \frac{2S'(R)}{T'(R)}.$$

Let $\xi = h^{-1}$ be the inverse map on an attracting branch on which $h'(R) < 0$. If the fast transient reaches this branch, then the reduced slow flow satisfies

$$\frac{d\widehat{a}_{\tau_s}}{d\tau_s} = \frac{\alpha(\xi(\widehat{a}_{\tau_s}))}{\Lambda_s T'(\xi(\widehat{a}_{\tau_s}))},$$

and, equivalently, with $\widehat{a}_{\tau_s} = h(\widehat{R}_{\tau_s})$,

$$\frac{d\widehat{R}_{\tau_s}}{d\tau_s} = \frac{\alpha(\widehat{R}_{\tau_s})}{\Lambda_s T'(\widehat{R}_{\tau_s}) h'(\widehat{R}_{\tau_s})}. \tag{28}$$

Since $T'(R) > 0$ for small positive $R$ and the attracting branch satisfies $h'(R) < 0$, the sign of the reduced $R$-flow is opposite to the sign of $\alpha(R)$. Thus, on such a branch, $\alpha(R) > 0$ drives the alignment $R$ toward smaller values and may lead to feature unlearning, whereas $\alpha(R) < 0$ drives $R$ away from the origin and supports feature learning. In this sense, feature unlearning requires both the existence of an unlearning branch and an initialization that places the trajectory on it after the fast transient.

**Case 1: The unlearning branch exists but is not reached.** Suppose that there exists $R_{\mathrm{u}} > 0$ such that $\alpha(R) > 0$ for $R \in (0, R_{\mathrm{u}})$ and $\alpha(R_{\mathrm{u}}) = 0$. Then the component of $\mathcal{S}_0^+$ corresponding to $R \in (0, R_{\mathrm{u}})$ is the unlearning branch: once the trajectory is placed on this component, the reduced slow flow moves toward $R = 0$ and $a = \infty$. However, the branch selected after the fast transient is controlled by the initial second-layer scale $\bar{a}$. More precisely, if $R_{\mathrm{in}} = h^{-1}(\bar{a})$ is the point reached by the fast equation with $a$ frozen at $\bar{a}$, then the reduced slow dynamics starts from $R_{\mathrm{in}}$ on the critical manifold. In this case, we do not analyze possible jumps between distinct attracting components of the critical manifold after this fast relaxation. Motivated by our numerical simulations, where such inter-branch jumps were not observed, we assume that the slow trajectory remains on the attracting component selected by the initial fast flow.

Consequently, under this branch-preservation assumption, even when the unlearning branch exists, feature unlearning does not occur if $R_{\mathrm{in}}$ lies outside this component. For example, if $R_{\mathrm{in}} > R_{\mathrm{u}}$ and the forward slow orbit remains in a region where $\alpha(R) < 0$ and $h'(R) < 0$, then (28) gives $d\widehat{R}_{\tau_s}/d\tau_s > 0$, so the trajectory moves away from the origin. If this orbit converges to a finite equilibrium $R_{\mathrm{FL}} > 0$ on the critical manifold, then

$$\lim_{\tau_s \to \infty} \widehat{R}_{\tau_s} = R_{\mathrm{FL}} > 0, \qquad \lim_{\tau_s \to \infty} \widehat{a}_{\tau_s} = h(R_{\mathrm{FL}}) < \infty,$$

and hence the system achieves feature learning in the sense of Definition 1. This case explains why changing only $\bar{a}$ can switch the outcome: since $h(R) = (c_{\star,1}/c_1)R^{-1} + o(R^{-1})$ as $R \to +0$, larger values of $\bar{a}$ place the fast-flow endpoint closer to the origin, while smaller values of $\bar{a}$ may place it on a different branch where the slow flow does not unlearn the feature.

**Case 2: The unlearning branch is absent.** A more fundamental obstruction occurs when the positive branch adjacent to $R = 0$ is not an unlearning branch at all. In the present fast–slow mechanism, convergence to $R = 0$ along $\mathcal{S}_0^+$ requires $\alpha(R) > 0$ for all sufficiently small positive $R$. If, instead, there exists $\rho > 0$ such that

$$\alpha(R) < 0, \qquad R \in (0, \rho), \tag{29}$$

then (28) implies $d\widehat{R}_{\tau_s}/d\tau_s > 0$ near the origin. Hence $R = 0$ is repelling for the reduced slow flow on the attracting positive branch, and no choice of $\bar{a}$ that lands on this local branch can yield feature unlearning through the mechanism analyzed in Theorem 2. In this case, the unlearning branch itself is absent.

The condition (29) can be read directly from the coefficient expansion of Lemma 10. For instance, if $k_0 + 1 < 2k_1$ and $c_{\star,k_0} c_{k_0} > 0$, then the leading term of $\alpha(R)$ is negative, and therefore $\alpha(R) < 0$ for all sufficiently small $R > 0$. In the borderline case $k_0 + 1 = 2k_1$, the same conclusion holds whenever the leading coefficient

$$-2(k_0 - 1)k_0! c_{\star,k_0} c_{k_0} c_1^2 + 2(k_1 - 1)k_1! c_{\star,1} c_1 c_{k_1}^2$$

is negative; if this coefficient vanishes, the sign is determined by the first non-zero higher-order term in the expansion of $\alpha$. Thus, the coefficients of the teacher and student nonlinearities determine whether an unlearning branch exists, while the initial value $\bar{a}$ determines which branch is selected. Feature unlearning requires both conditions; if either one fails, the fast–slow dynamics leads to feature learning, with the alignment bounded away from zero.

## G. Additional simulation

We perform numerical simulations of the ODE (6) for multiple choices of activation and link function coefficients $(c_k, c_{\star,k})$. In particular, we consider the case with $\bar{k}_\star = \bar{k} = 7$ and fix some coefficients as $c_1 = c_2 = c_3 = 1$ and $c_{\star,1} = 1$. Then, we vary the remaining coefficients as $c_{\star,2}, c_{\star,3} \in \{-5, -1.67, 1.67, 5\}$.

Figure 9 shows the results. In all cases satisfying the initialization condition in Assumption 5 identified in the theoretical analysis, we consistently observe that the dynamics of $(R_\tau, a_\tau)$ follows the result in Section 4. Consequently, these results validate the overview of the feature unlearning along attracting branches of the critical manifold.

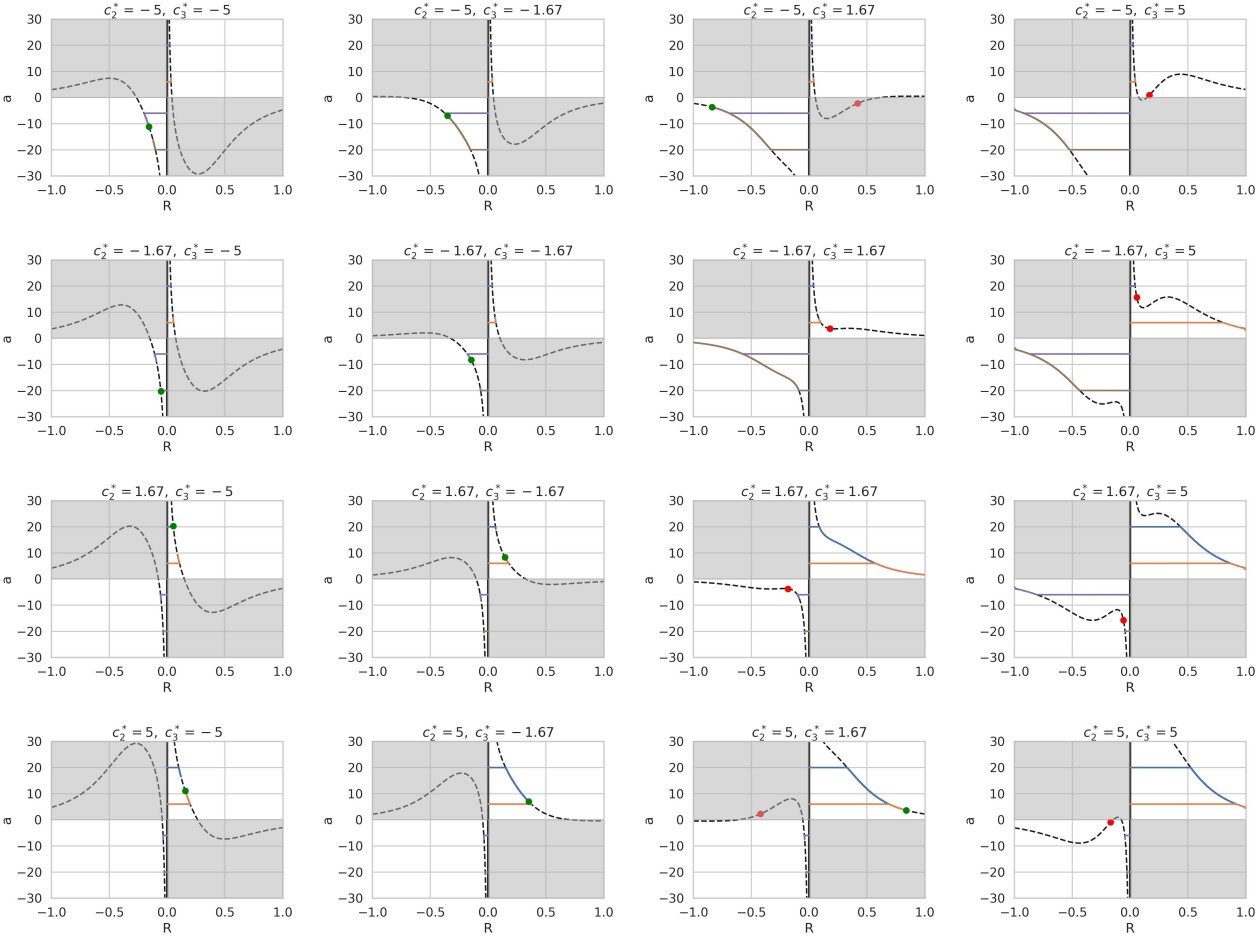

*Figure 9.* Dynamics of $(R_\tau, a_\tau)$ by the ODE (6) with the various coefficients of the link function and the activation function.

