# OpenReview forum: "Dichotomy of Feature Learning and Unlearning: Fast-Slow Analysis on Neural Networks with Stochastic Gradient Descent"
_ICML.cc/2026/Conference — ICML 2026 regular_

### Official Review · Reviewer_i1aM · 2026-03-08

**Soundness:** 3
**Presentation:** 3
**Significance:** 3
**Originality:** 3
**Overall Recommendation:** 5
**Confidence:** 3

**Summary:**

This paper investigates the dichotomy of feature learning and feature unlearning in two-layer neural networks trained with one-pass SGD.  The authors derive a deterministic, continuous-time PDE in the infinite-width limit to describe the network's macroscopic variables. The central observation is the "fast-slow" dynamic: the alignment of first-layer weights develops rapidly, the second layer weights develop slowly, and the latter decides whether feature unlearning occurs.

**Compliance With Llm Reviewing Policy:**

Affirmed.

**Final Justification:**

This is a good work and I maintain my positive recommendation for it.

**Key Questions For Authors:**

1. Could the authors clarify how the mechanism of feature unlearning in this specific setting differs from related phenomena in the literature? Specifically, Montanari & Urbani (2025) link feature unlearning to overfitting. However, this paper utilizes online, one-pass SGD where fresh samples are drawn at each step , meaning the test loss continues to decrease. Furthermore, the authors claim that the progressive loss of alignment means the network eventually learns in a "lazy regime". Could the authors reconcile this terminology, given that strict lazy training implies that feature alignment barely moves from initialization, rather than experiencing a transient spike followed by a decay?

2. Could you elaborate on the necessity of Assumption 6 and Assumption 7? Are these conditions an absolute mathematical requirement for feature unlearning, or are they primarily technical conveniences designed to close the proofs?

3. Could the authors discuss what will happen if the initialization is not symmetric? If the second layer uses a standard Gaussian initialization instead, does the fast-slow dynamics still exist?

4. While the fast-slow ansatz is largely motivated by numerical observations, is it possible to rigorously prove it across the entire training trajectory? Given that Eq. (6) is a relatively simple two-dimensional ODE, it seems plausible that formal conditions for this ansatz could be established. Do the authors expect this to be provable, and what mathematical hurdles prevent a full proof in the current work?

**Limitations:**

The authors should briefly discuss what will happen if the assumptions break.

**Strengths And Weaknesses:**

Strength: The fast-slow dynamics of SGD observed by this paper is novel and contains valuable insights to the theoretical machine learning community. The theory is elegant, the derivations are sound, and the results are well-supported by experiments.

Weakness: The results of this paper rely on several technical assumptions. The initialization (Assumption 5) is artificial and deviates from standard random initialization practices. The separation of time scales is only an ansatz supported by experiments. Assumption 6 also looks a bit artificial and it is difficult to understand why Assumption 6 is essential to feature unlearning.

---

> ### Author Rebuttal · Authors · 2026-03-29
>
> Thank you very much for your thoughtful comments.
>
> ---
>
> **Q1**: How the mechanism of feature unlearning in this specific setting differs from related phenomena in the literature? /  Could the authors reconcile this terminology "lazy regime".
>
> **A**: Thank you for this important question.
>
> First, our results essentially characterize feature unlearning through **the population dynamics of two-layer neural networks**, without depending on properties of training data. This is fundamentally different from existing results (e.g., Montanari & Urbani (2025)), as our description is not tied to finite-dataset memorization or the associated notion of overfitting. This distinction stems from the fact that we study one-pass online SGD with fresh samples (and additional support is provided by the fact that, in Appendix Section C, we independently derive the fast–slow ODE from the population loss). In summary, our key novelty lies in showing that fast–slow dynamics intrinsic to two-layer neural networks can induce feature unlearning, independently of overfitting to the training data.
>
> Second, regarding the term *lazy regime*, we agree that the current wording may be misleading. The learning trajectory we describe is not a strict lazy regime; rather, it is characterized by an initial phase where alignment is clearly acquired (i.e., feature learning), followed by a long-time limit in which the learned features are lost (eventually leading to a state that coincides with the lazy regime). Since the dynamics depart from the lazy regime during training, the behavior is qualitatively different from the standard notion of the lazy regime.
>
> Both of these points are essential, and we will revise the introduction and the relevant sections to clarify them explicitly.
>
> ---
>
> **Q2**: Could you elaborate on the necessity of Assumption 6 and Assumption 7?
>
> **A**: Regarding the necessity of Assumptions 6 and 7, we view them as follows. The essential requirement is the sign condition on $\alpha(R)$, which determines the direction of the flow on a manifold and also whether feature unlearning appears.
>
> Assumption 6 provides an explicit sufficient condition that allows $\alpha(R)$ to be directly verified from the Hermite coefficients of the teacher and student. Therefore, we do not claim that it constitutes the most general necessary and sufficient condition.
>
> Assumption 7 is introduced to ensure that the initial value $\bar{a}$ lies in the basin of attraction of the branch of unlearning that $\alpha(R)$ has a specific sign. This, too, is a sufficient condition adopted to enable a clean theorem statement.
>
> Therefore, while these conditions tend to be formal in nature, they play an essential role in determining whether unlearning occurs based on the sign of $\alpha(R)$. We will add this explanation to the main text to provide further clarification.
>
> ---
>
> **Q3**: Could the authors discuss what will happen if the initialization is not symmetric?
>
> **A**: For asymmetric initialization, particularly when standard Gaussian initialization is used in the second layer, the current two-dimensional ODE no longer holds as is. The reason is that exchangeability breaks down, and one must track each neuron’s $(a_i, R_i)$ individually, or equivalently, track their distribution. In particular, in the zero-mean Gaussian case, cancellation between positive and negative $a_i$ can significantly alter the initial drift. Therefore, we do not claim that the same theorem applies directly in this setting. However, the fast–slow effect at the population level may still persist, and we consider this an important direction for future investigation.
>
> ---
>
> **Q4**: While the fast-slow ansatz is largely motivated by numerical observations, is it possible to rigorously prove it across the entire training trajectory?
>
> **A**: We believe that a direct geometric proof is possible without explicitly relying on the fast–slow ansatz. Specifically, in our setting, one can exploit the following properties: (i) the trajectory remains in the first quadrant; (ii) as unlearning progresses, the spectral gap between the fast and slow modes becomes increasingly pronounced; and (iii) the trajectory cannot cross the R-nullcline in the horizontal direction. Using these observations, one can argue that the trajectory is constrained to the relevant branch and converges toward R \to 0.
>
> In summary, while the fast–slow decomposition is adopted in this paper as the most transparent way to describe the phenomenon, the essential mechanism underlying the unlearning case is likely to admit a more direct interpretation in terms of trajectory geometry and nullcline structure.

---

> > ### Author Rebuttal · Reviewer_i1aM · 2026-04-01
> >
> > The authors have  addressed all of my concerns. I am pleased to maintain my positive recommendation for this work.

---

### Official Review · Reviewer_FWuY · 2026-03-10

**Soundness:** 2
**Presentation:** 2
**Significance:** 3
**Originality:** 2
**Overall Recommendation:** 4
**Confidence:** 3

**Summary:**

This paper studies the learning dynamics of a two-layer network trained with SGD to study the mechanism behind feature unlearning. This paper uses the Tensor Programs framework to derive deterministic ODEs describing the dynamics. They describe the emergence of fast-slow dynamics and propose it as a mechanism that can explain feature unlearning as slow dynamics on the manifold.

**Compliance With Llm Reviewing Policy:**

Affirmed.

**Final Justification:**

I appreciate the authors’ thoughtful and comprehensive response. I have raised my score as my concerns have been addressed.

**Key Questions For Authors:**

I have a few questions for the authors.

1. Based on the comments above, could you better situate your paper in the literature? In particular, highlight the point of difference with Montanari 2025?
2.  Is there any other evidence of unlearning in practice?
3. How does unlearning impact the generalisation error and training error?
4. What determines which point on the manifold the networks reach on the fast time scale ? (initialisation? (Have you tried to also vary the first layer weight norm?) Teacher parameters?)
5. How does feature unlearning relate to specialisation in the Saad and Solla style analysis? ( I understand there are slightly different theoretical settings, but are there any analogies/links?)
6. How does the fast inital alignment relate to the silent alignment effect, the saddle-to-saddle literature (Atanasov et al. 2021, Jacot et al., 2021)? (Here again, I understand there are slightly different theoretical settings, but are there any analogies/links?)

I look forward to your answer and remain open to increasing my score upon clarification of the contributions and limitations.

**Limitations:**

The paper does not point out any limitations of the method. Multiple limitations should be highlighted. The method is set in two-layer networks, which are far from real networks. The method also makes several limiting assumptions (e.g., both the batch size n and the feature dimension d diverge to infinity while preserving n/d or an infinite width). Even though they are important and common assumptions in this style of analysis, they should be clearly pointed out. The author does not propose any furtherwork.

**Strengths And Weaknesses:**

I would like to first thank the author for their effort and for this interesting work.

**Soundness**

**Strengths:** The paper is technically sound, and the claims are well supported by analysis.

**Presentation**

**Strengths:** The paper is well written. The figures help the understanding, and are clear.

**Weaknesses:** For clarity, the paper should introduce the definition of feature unlearning and a short explanation of the fast-answer-slow time scale earlier. Furthermore, one downside is that the paper is not well-positioned within the current literature, which creates clarity issues. I discuss this more in the significance park.

**Significance**

**Strengths:** The paper is interesting, timely, and highly relevant to the ICML community. Understanding feature learning is crucial to the entire field as AI advances rapidly.

**Weaknesses:** I point to literature which I believe could provide more context.

1. **Dynamics of Linear Network + Dynamical Feedback Principle:** The paper fails to cite important parts of the literature. Saxe et al. (2014) pioneered the dynamical approach to describing the dynamics of linear networks. Even though this paper is in a non-linear setting, this reference is crucial here. In recent years, these dynamical methods have been extended to understand multiple phenomena, demonstrating their versatility (Nam et al., 2025 (review); Pinson et al., 2023; Simon et al., 2023; Jarvis et al., 2023; Zhang et al., 2024a). In particular, the phenomenon close to the fast-slow dynamics has been studied in linear networks (i.e., Domine et al. 2025, Nam et al. 2025, Azulay et al. 2021) and non-linear networks from the perspective of balanced initialisation ( relative scaling of a with respect to the first layer)  (Kunin et al. 2025). This phenomenon has previously been linked to the feature-learning literature. The phenomena studied in this paper are closely related to those in Kunin et al. (2025), tho in a different setup.

2. **Dynamics of nonlinear networks.** Another line of work has examined the dynamics of two-layer networks, focusing on order parameters (Saad and Solla, 1995). This literature is partially cited in this paper (Gold et al 2019). However, extensions of this setting are not cited. In particular, the paper by Jarvis et al. (2025) studies the imbalanced initialisation setting (modifying the norm of a with respect to w) and studies the rich-and-lazy dynamics using the specialisation measure (alignment with the teacher).

3.  **Rich and Lazy learning.** The literature on the rich-and-lazy learning regime is also lacking. Critical papers: (Jacot et al., 2018; Chizat et al., 2018; Jacot et al., 2021) should be cited.  In general, feature learning is defined as a non-static NTK. The feature unlearning still probably has a non-static NTK. Therefore, it might still be technically feature learning. My understanding of the feature-unlearning phenomenon is closer to the term feature of unalignment.

I understand that these papers are not all directly in the paper's subfield, but I believe they provide context and are crucial to the paper.
The contribution of the paper are not clear due to the lack of literature review and limitations ( discussed bellow)

**Originality**

**Strengths:** This paper offers links between the fast/low dynamics and feature un-learning. The setting and method used to study this phenomenon are new here. The paper show results in practical settings.

---

> ### Author Rebuttal · Authors · 2026-03-29
>
> **C**: The phenomenon may be closer to feature unalignment than feature unlearning.
>
> **A**: We agree. In this paper, “feature unlearning” does not mean whether the NTK is static; rather, it refers to the more specific phenomenon that alignment with the teacher direction first increases and later decays. We will clarify this point in the introduction and explicitly note that the phenomenon may also be interpreted as feature unalignment.
>
> ---
>
> **Q1**: Positioning in the literature, especially relative to Montanari & Urbani (2025)
>
> **A**: We will clarify the position of our work along three lines: (i) dynamics of linear networks, (ii) order-parameter analysis in teacher–student models, and (iii) NTK / lazy–rich / small-initialization theory.
>
> First, our work belongs broadly to the tradition of describing learning dynamics using a few macroscopic variables, following works such as Saxe+(2014) and later developments.
>
> Second, it is related to studies on initialization and lazy–rich transitions, including Azulay+(2021), Dominé+(2025), and Kunin+(2024, 2025). While those works ask how initialization or layerwise scaling induces lazy/rich behavior or staircase-like feature learning, our question is different: *why learned alignment disappears at long times*, and we provide a geometric explanation of this effect.
>
> Third, our paper is closely related to nonlinear two-layer teacher–student analyses such as Saad & Solla (1995), Goldt+(2019), and Jarvis+(2025). Our setting is distinguished by a single-index teacher and symmetric initialization, under which the dynamics close in two variables $(R,a)$. This shifts the focus from unit-wise specialization to the global retention and loss of alignment.
>
> Among prior works, Montanari & Urbani (2025) is the closest to ours. The key difference is that **they study feature unlearning associated with dataset-dependent overfitting/decoupling, whereas we analyze population dynamics and identify a geometric mechanism of unalignment independent of a finite training set**. Methodologically, they use gradient flow + DMFT, whereas we use online one-pass SGD + Tensor Programs + fast–slow analysis.
>
> ---
>
> **Q2**: Is there evidence of unlearning in real networks?
>
> **A**: Yes. In addition to our numerical result (shown in Figure 7) and Montanari & Urbani (2025), some studies (for example, Vyas+(2023 ICLR), Gupta+(2026 TMLR)) investidates neural networks with real data and shows limitations of feature alignment based on NTK's evolution, as well as how the learned representations in each layer are lost.
>
> ---
>
> **Q3**: How does unlearning affect generalization and training error?
>
> **A**: Our focus is not the train/test gap for a fixed dataset, but the dynamics of population/test loss, since our setting uses fresh samples at every step and the ODE is derived in a large-batch limit. Thus, our main result is that population/test loss does not decrease effectively because of feature unalignment, and we cannot investigate the training error and overfitting.
>
> ---
>
> **Q4**: What determines where the trajectory reaches on the manifold?
>
> **A**: The arrival point is determined by the intersection of the line $a=\bar a$ and the stable branch $a=h(R)=\frac{2S'(R)}{T'(R)}$. Hence it depends on (i) the initial second-layer scale $\bar a$, and (ii) the Hermite coefficients $c_{\star,k},c_k$ through $S$ and $T$. The direction of the later slow flow is determined by $\alpha(R)=S(R)T'(R)-2S'(R)T(R)$, so these coefficients govern both where the trajectory lands and whether it later moves toward learning or unlearning.
>
> ---
>
> **Q5**: Relation to Saad–Solla-type specialization
>
> **A**: Our setting uses balanced initialization, so all neurons remain equivalent, and the kind of specialization studied by Saad & Solla (1995), Goldt+(2019), and Jarvis+(2025) does not occur.
>
> ---
>
> **Q6**: Relation to silent alignment / saddle-to-saddle dynamics
>
> **A**: The early phase is similar to silent alignment, but the late-time interpretation differs. As shown in Appendix D, $g(0,\bar a)=0$ and $f(0,\bar a)>0$, so initially a is nearly frozen while $R$ grows rapidly. This resembles the silent alignment by Atanasov+(2021). The difference is that, in our setting, the alignment acquired early can later decay toward $R\to 0$ because of slow drift.
>
> There is also a possible connection to the saddle-to-saddle. In our current setting, the manifold geometry is simple, so multistage behavior does not appear. But for other Hermite coefficients, the manifold may become more complex, and the same fast–slow mechanism could potentially generate multistage behavior. We do not pursue this here because of technically nontriviality, but we view it as an important direction for future work.
>
> ---
>
> All of these points are important, and we will reflect them in the revised manuscript. We sincerely thank you for your thoughtful suggestions and insightful questions, which have helped us clarify both the positioning and the core contributions of the paper.

---

> > ### Author Rebuttal · Reviewer_FWuY · 2026-04-02
> >
> > I appreciate the authors’ thoughtful and comprehensive response. I have raised my score as my concerns have been addressed.

---

### Official Review · Reviewer_tCJY · 2026-03-10

**Soundness:** 3
**Presentation:** 3
**Significance:** 2
**Originality:** 2
**Overall Recommendation:** 5
**Confidence:** 4

**Summary:**

This paper studies the optimization dynamics of a two layer neural network on a single-index model with isotropic Gaussian inputs in the mean field regime. As a result of their initialization, they can derive a 2-dimensional ODE which governs the dynamics of two order parameters (the alignment with the target direction $R$ and the output scale $a$) and they observe that this 2-dimensional ODE exhibits two-timescale dynamics. They then explicitly make this two-timescale ansatz to analyze the resulting 2-dimensional dynamical system and show that some initial conditions exhibit "feature unlearning" in which $R$ grows at first before converging to $0$.

**Compliance With Llm Reviewing Policy:**

Affirmed.

**Final Justification:**

The authors have addressed my concerns and I have updated my score accordingly.

**Key Questions For Authors:**

- How does your notion of feature unlearning relate to that of Montanari & Urbani (2025)? My understanding from that paper was that feature unlearning was a result of having a finite dataset that you eventually memorize, and that this memorization causes growth in $a$ and leads to the feature unlearning regime. However, this paper effectively studies the population dynamics and the feature unlearning is due to a failure of gradient flow to coordinate $(R,a)$?
- What are the challenges in directly analyzing the dynamics in equation (6) without the two-timescale ansatz?
- Without introducing the $\epsilon$ parameter, are there any natural limits under which the two-timescale ansatz is exact?

**Limitations:**

yes

**Strengths And Weaknesses:**

Strengths:
- The theory in this paper is well complemented by the synthetic experiments and it is clear from the $(R,a)$ plots that the two-timescale formalism is the right approach for this problem.
- The intuitive explanation for feature unlearning and the diagrams throughout the paper are very clear.
- The paper gives explicit sufficient conditions for feature unlearning as a function of the Hermite coefficients of the learner and target activations.

Weaknesses:
- The specific set of limits taken in the paper is non-standard:
  - Despite using a mean-field network, the paper does not scale $\gamma$ with $m$, so that the network dynamics are degenerate on $O(1)$ timescales. This is resolved in the later limits by taking $\tau \propto t/m$
  - As a result of the above point, the learning rate is tiny for this network when the width is large. The width therefore plays a hidden dual role of making the step size small enough for the continuous-time gradient flow limit to hold. This should be broken into two assumptions: one about the width diverging and another about the effective step size going to $0$.
  - The paper fixes a ratio $n/d$ where $n$ is the batch size before taking $m$ to infinity. As a result, the total number of samples seen during training is $n \times t \propto n \times m \times \tau$. Since $n \propto d$, the learner sees $\Theta(md)$ samples. This is far more than is necessary to concentrate to the population dynamics. For example, the analysis in [1] showed that $\Theta(d)$ samples suffice.
- Related to the non-standard limit point above, the "width" of the network plays no role in this paper because of the symmetric initialization and constraining $\tau = O(1)$. As a result, the paper is really about learning a single neuron teacher $\sigma_\star$ with a single neuron learner $f = a \sigma(w \cdot x)$.
- Despite the apparently simplicity of the system, the paper does not actually directly analyze the 2-dimensional dynamics of $(R,a)$ and instead resorts to the two-timescale ansatz.

[1] The merged-staircase property: a necessary and nearly sufficient condition for SGD learning of sparse functions on two-layer neural networks

Minor typos:
- In Assumption 2, $w^\star$ should be initialized as $\mathrm{Unif}(S^{d-1}(\sqrt{d}))$ to be consistent with the rest of the paper. Otherwise the Hermite expansions of $\sigma_\star$ are not valid as $\langle w_\star, x \rangle/\sqrt{d}$ is not $N(0,1)$. However, this normalization can safely be absorbed into $\sigma_\star$ if you condition on the norm (e.g. define $\tilde \sigma_\star(z) := \sigma_\star\left(z \frac{\sqrt{d}}{\|w_\star\|}\right)$).
- The derivation in Appendix C is incorrect as written. For example, the statement that $Q_{ij} = Q$ with the symmetric initialization only holds in the limit as $d \to \infty$ and does not hold for finite $d$, but this limit is not explicitly taken in Appendix C.

---

> ### Author Rebuttal · Authors · 2026-03-29
>
> Thank you very much for your thoughtful comments
>
> ---
>
> **W1**: The specific set of limits taken in the paper is non-standard
>
> **A**: Indeed, the limit considered in this paper is not identical to the standard mean-field scaling. However, the key point is that **the fast–slow dynamics observed in this work are NOT an artificial phenomenon that depends solely on this setup**. In fact, as shown in Section C in the appendix, a similar fast–slow structure arises by investigating the dynamics of general population loss. Therefore, it reveals that the fast–slow dynamics we identify are not merely artifacts of a specific discretization or finite-width noise, but rather constitute a more general geometric phenomenon shared across a broader class of learning dynamics.
>
> Also, about the large $m$ regime, while it is true that the width $m$ increases to infinity in theory, we have shown that phenomena consistent with theory are observed even finite $m = 500$ in experiments in Section 6.
>
> ---
>
> **W2**: the width plays no role because of the symmetric initialization and constraining $\tau = O(1)$. As a result, the paper is really about learning a single neuron
>
> **A**: **Being reduced to a two-dimensional system at the level of order parameters is not equivalent to being a single-neuron model**. In our derived ODE, the closure $Q = R^2$ and the term $T(R)$ **arise from the averaging over many neurons** as $m \to \infty$. This is fundamentally different from the dynamics of a single-neuron model. Even though the width $m$ does not appear explicitly in the ODE, it remains an essential parameter underpinning the validity of the limit.
>
> ---
>
> **W3**: the paper does not directly analyze the dynamics of $(R,a)$ and instead resorts to the two-timescale ansatz.
>
> **A**: As you pointed out, the main analysis in this paper employs a fast–slow decomposition. However, this should not be interpreted as a lack of direct understanding of the two-dimensional system; rather, **we adopt the fast–slow framework as the most natural tool for transparently describing the long-time behavior.**
>
> In fact, at least for the unlearning case, we believe that much of the behavior can be established directly from the geometric properties of the trajectories, even without explicitly invoking the fast–slow decomposition. More specifically, by combining (i) the fact that the trajectory remains in the first quadrant, (ii) the observation that the eigenvalue scale separation is further amplified as unlearning progresses, and (iii) the fact that the trajectory cannot cross the $R$-nullcline in the horizontal direction, one can argue directly that the trajectory drifts toward $R \to 0$ along the relevant branch.
>
> ---
>
> **Q1**: How does your notion of feature unlearning relate to that of Montanari & Urbani (2025)?
>
> **A**: First, a key similarity is that both studies argue that alignment decay occurs. In contrast, a key difference is that while the Montanari & Urbani investigates the relationship between unlearning and overfitting caused by the influence of training data, we examine population dynamics that are independent of the training data. This is because our analysis focuses on a one-pass SGD setting, where new samples arrive at each step, and examines large batch sizes. Consequently, our analysis does not examine overfitting (the gap between training and test errors) as in Montanari & Urbani; instead, we analyze population-level phenomena and provide a geometric interpretation of unlearning:after a fast attraction to the critical manifold, the direction of the slow flow is determined by the sign of $\alpha(R) = S(R)T'(R) - 2S'(R)T(R)$, which drives the system toward $R \to 0$ and $a \to \infty$.
>
> ---
>
> **Q2**: Directly analyzing the dynamics without the two-timescale ansatz?
>
> **A**: The main difficulty in directly analyzing Equation (6) without invoking the two-time-scale hypothesis lies not so much in the local vector field itself, but in achieving global geometric control over the entire trajectory. Specifically, when using sets that give rise to qualitatively different flows, such as the manifold in this study, it becomes difficult to rigorously characterize the properties of the trajectory.
>
> ---
>
> **Q3**: Without $\varepsilon$, are there any natural limits under which the two-timescale ansatz is exact?
>
> **A**: We understand your question as asking whether there exists a natural limiting ODE, without introducing a parameter $\varepsilon$, for which the two-time-scale hypothesis holds rigorously.
>
> In short, we believe that it is difficult at present to identify a single natural limit under which the two-time-scale hypothesis holds rigorously over the entire trajectory. In particular, the ODE limit we consider (i.e., $m \to \infty$ with $\tau \propto t/m$) is introduced to obtain a closed macroscopic dynamics, and it does not automatically imply a uniform separation of time scales.

---

> > ### Author Rebuttal · Reviewer_tCJY · 2026-03-31
> >
> > **W1** My concern is about the presentation of the setting. As far as I can tell, the paper is in a standard mean field setting, but then uses the knob $m \to \infty$ to control two separate approximations: (1) mean field fluctuations and (2) the gradient flow approximation. In the mean field/tensor programs literature these are generally separated by using a learning rate of $\gamma = \gamma_0 m$, taking $m \to \infty$ to control the mean field fluctuations, and then taking $\gamma_0 \to 0$ to couple to gradient flow. This makes clear that the continuous-time dynamics are an additional approximation on top of the mean field limit, rather than an automatic consequence of width as claimed in Section 3.1. I also raised the issue of sample complexity: It seems that this paper requires far more samples to concentrate to the population dynamics than are actually necessary.
> >
> > **W2** Thank you for clarifying the misunderstanding. I agree that the width plays a nontrivial role.
> >
> > **W3** As I mentioned in my review it's clear from the plots that the two-timescale dynamics are natural for this problem, but as far as I can tell there is no theoretical justification in the paper for them. If the logic in "In fact, at least for the unlearning case [...] along the relevant branch." could be formalized, it would be a strong addition to the paper.
> >
> > **Q1-Q3** I thank the authors for their responses, and these questions have been clarified for me.

---

> > > ### Author Response · Authors · 2026-04-02
> > >
> > > Thank you very much for your thoughtful comments.
> > >
> > > ---
> > >
> > > **W1**: The paper is in a standard mean field setting, but then uses the knob $m \to \infty$ to control two separate approximations. / This paper requires far more samples to concentrate to the population dynamics than are actually necessary.
> > >
> > > **A**: Thank you for this important comment. We agree that distinguishing the two limits ($m \to \infty$ and $\gamma_0 \to 0$) more explicitly would improve clarity. This makes clear that the continuous-time dynamics arise as an additional approximation, rather than as a direct consequence of width alone. We will revise Section 3.1 and the derivation in the appendix to reflect this separation more explicitly.
> > >
> > > Regarding the sample complexity, this rescaling leads to the sample complexity of $O(d)$: time $t$ corresponds to $t/\gamma_0$ gradient steps, consuming $tn/\gamma_0$ samples in total. This removes dependence on the width $m$, addressing the concern that our setting requires too many samples.
> > >
> > > ---
> > >
> > > **W3**: it's clear from the plots that the two-timescale dynamics are natural for this problem, but as far as I can tell there is no theoretical justification in the paper for them
> > >
> > > **A**: In Section D, we provide a brief theoretical analysis of the causes of fast-slow dynamics arising from the derived ODE.
> > > The results consist primarily of two claims.
> > > First, at time zero (when, by definition, $R_\tau$ is also zero), $da_\tau /d \tau = 0$ and $d R_\tau / d \tau > 0$ hold strictly. This strictly demonstrates that at the initial time, a is stationary and only $R_\tau$ evolves.
> > > Second, for a short time after $R_\tau$ has begun to grow, we show that $d a_\tau / d R_\tau = O(R_\tau)$. This demonstrates that when $R_\tau$ is growing slowly (and has not yet reached the manifold), the relative growth of $a_\tau$ is suppressed.
> > >
> > >
> > > **--additional reply--**
> > >
> > > We have realized that feature unlearning under Assumptions 6 and 7 can be established by directly analyzing the ODE, **without** invoking the fast-slow ansatz. Specifically, the trajectory remains confined to a region in the (R,a) plane bounded by the a-axis and the relevant nullcline, within which da/dt > 0. This implies monotonic growth of a and a \to \infty and R \to 0.
> > > We will include this result in the revised manuscript to further support both feature unlearning and the fast-slow interpretation.
> > >
> > >
> > > ---
> > >
> > > Thank you for your constructive feedback. Your comments have greatly improved the paper. Please let us know if you have any additional concerns or comments.

---

### Official Review · Reviewer_nEZv · 2026-03-11

**Soundness:** 3
**Presentation:** 3
**Significance:** 2
**Originality:** 3
**Overall Recommendation:** 3
**Confidence:** 3

**Summary:**

This paper introduces a concept of feature unlearning and considers learning a single-index teacher with an infinite-width limit two-layer neural network updated with a large-batch SGD. Then this paper observes and shows the dichotomy of feature learning and unlearning by utilizing the fast-slow dynamics.

**Compliance With Llm Reviewing Policy:**

Affirmed.

**Final Justification:**

I will maintain my current score, since relaxing the single-index assumption substantially strengthens the contribution.

**Key Questions For Authors:**

1. I am wondering the motivation for the definition of ODE (6). Is it a standard definition or other definitions also work?
2. This paper considers $m \rightarrow \infty$, I am wondering if there is a threshold for $m$ such that if $m$ is larger than the threshold, the feature unlearning will appear. What if $m=2$?
3. The teacher model in this paper is a single-index model, if considering more general case - multi-index teacher model, is it possible to show some similar results?

**Limitations:**

yes

**Strengths And Weaknesses:**

Strengths:
1. This paper is technically sound.
2. This paper is clearly written and well structured.
3. This paper does provide an interesting point to explain the overfitting in deep learning.

Weaknesses:
1. This paper has limited contribution. Although the phenomenon of feature unlearning is quite interesting and it can explain the overfitting partially, we don't want the feature unlearning in practice and the overfitting in deep learning is well-known.

---

> ### Author Rebuttal · Authors · 2026-03-29
>
> Thank you very much for your important comments:
>
> ---
>
> **W1**: We don't want the feature unlearning in practice and the overfitting in deep learning is well-known.
>
> **A**: The main objective of this paper is **NOT to present feature unlearning itself as a desirable phenomenon**, but rather to explicitly characterize the conditions under which feature unlearning occurs in an analytically tractable manner. In other words, within our framework, it is also **possible to construct sufficient conditions under which feature learning is preserved** by examining conditions that reverse the direction of the slow flow. Specifically, if $k_0 + 1 < 2 k_1$ and $c_{\star, k_0} c_{k_0}>0$ holds, we may show $\alpha(R)<0$ hold, hence we have $\alpha(R) < 0$ as $R \to 0$. Therefore, on the relevant attracting branch, the slow flow decreases $a$, and the corresponding $R = h^{-1}(a)$ increases. Consequently, feature unlearning does not occur.
>
> In summary, the contribution of this work is not merely to reconfirm the known and undesirable phenomenon of unlearning, but to identify an explicit criterion that separates feature learning from feature unlearning. We believe that this condition provides new insights into feature unlearning and helps to further elucidate overfitting, which still contains many unresolved aspects.
>
> To further clarify this point, we will additionally derive rigorous conditions under which feature unlearning does not occur and include them in the revised manuscript.
>
> ---
>
> **Q1**: I am wondering the motivation for the definition of ODE (6). Is it a standard definition or other definitions also work?
>
> **A**: Equation (6) is **NOT introduced in an ad hoc manner**; rather, **it is rigorously proved to be a continuous-time limiting equation derived from the setup** of one-pass SGD under our two-layer neural network setting. In other words, the limit of our neural network and its update dynamics can be represented by Equation (6) without imposing any ad hoc assumptions (this is formally established in Proposition 1). More specifically, since the change in the macroscopic variables per step is of order $O(1/m)$, introducing the rescaled time $\tau = \gamma t / m$ yields a limiting ODE, which can be shown to take the form of Equation (6). In addition, in Section C of the Appendix, we independently derive the same ODE from the population gradient perspective, further confirming that there is no arbitrariness in the formulation.
>
> While different parameterizations or time scalings may lead to different limiting equations, Equation (6) should be regarded as the canonical limit corresponding to the model considered in this paper.
>
> ---
>
> **Q2**: if m is larger than the threshold, the feature unlearning will appear? What if $m=2$.
>
> **A**: If $m$ is finite but greater than a certain value, the answer is Yes. While our theory is an asymptotic analysis in the limit $m \to \infty$, in the finite-width experiments presented in the current manuscript, we already observe behavior consistent with the theory, namely, fast–slow dynamics and feature unlearning, for $m = 500$ and $m = 1000$.
>
> In contrast, for extremely small widths such as $m = 2$, the two-dimensional ODE approximation based on exchangeability and self-averaging is generally not valid, and our theory cannot be directly applied in such settings.
>
> We recognize that analyzing feature unlearning at finite m is an important direction for future work. We will revise the manuscript to explicitly mention this point in the conclusion section.
>
> ---
>
> **Q3**: if considering more general case - multi-index teacher model, is it possible to show some similar results?
>
> **A**: We agree that extending the analysis to the multi-index teacher setting is an important direction. However, at this stage, we cannot assert that the results for the single-index case generalize directly. In the multi-index setting, one needs to track multiple overlaps simultaneously rather than a single alignment parameter $R$, and the dynamics become inherently more complex due to competition and specialization across modes.
>
> Therefore, while it is an interesting possibility that similar results may hold, we cannot claim that the current analysis can be extended in a straightforward manner. That said, it is clear that this constitutes an important direction for future research, and we will revise the manuscript to mention this point explicitly in the conclusion section.

---

> > ### Author Rebuttal · Reviewer_nEZv · 2026-04-01
> >
> > Thank you for your detailed response. I have a follow-up question about W1:
> >
> > Some prior works, such as Zarifis et al. (2024), have already established convergence results for gradient descent when a single neuron is used to learn a single-index teacher model. In light of this, it remains unclear why it is necessary to study the single-index setting with a large number of neurons $m$, as well as the additional conditions required to prevent feature unlearning. More broadly, I am concerned that the single-index teacher assumption is rather restrictive, which may limit the overall scope and significance of the contribution.
> >
> > Zarifis, N., Wang, P., Diakonikolas, I., and Diakonikolas, J. Robustly learning single-index models via alignment sharpness. In Proceedings of the 41st International Conference on Machine Learning, volume 235, pp. 58197–58243. PMLR, 2024.

---

> > > ### Author Response · Authors · 2026-04-02
> > >
> > > Thank you very much for your important question. We’ll break our response down into a few points below.
> > >
> > > ---
> > >
> > > **W1-1**: Zarifis et al. (2024) have already established convergence results for gradient descent when a single neuron is used / why it is necessary to study the single-index teacher model
> > >
> > > **A**: First and foremost, we gently point out that **the approach and the goal of Zarifis et al. (2024) differ significantly from those of our study**.
> > > Since Zarifis et al. (2024) rely on inequality-based analysis for the required sample size and the number of iterations of SGD, this is essentially different from our analysis using ODEs to precisely identify the behaviour of SGD and multiple time scales to explain feature unlearning. In other words, it may be difficult for the approach of Zarifis et al. (2024) to find the fast-slow behavior during the training process we studied. Therefore, since the objectives of the analyses are fundamentally different, the existence of Zarifis et al. (2024) does not mean there is no reason to conduct single-index analysis.
> > >
> > > ---
> > >
> > > **W1-2**: it remains unclear why it is necessary to study the single-index setting with a large number of neurons $m$, as well as the additional conditions required to prevent feature unlearning
> > >
> > > **A**: In our derivation of ODE (Eq.6), several important coordinates, like the simplification identity $Q=R^2$ (Lemma 7) and the term $T(R)$, **arise from the averaging over many neurons** as $m \to \infty$. We can describe the macroscopic properties of neural networks by taking the average of many neurons, and this is fundamentally different from the dynamics of a single-neuron model. Even though the width $m$ does not appear explicitly in the ODE, it remains an essential parameter underpinning the validity of the limit.
> > >
> > > We also note that, our objective is not in designing an efficient algorithm to learn a single-index model, but rather in understanding how wide neural networks perform feature learning in a simple setting.
> > >
> > > ---
> > >
> > > **W1-3**: I am concerned that the single-index teacher assumption is rather restrictive, which may limit the overall scope and significance
> > >
> > > **A**: We also agree that the single-index teacher assumption is indeed simplistic, but the single-index teacher serves as a canonical and well-established testbed for analyzing feature learning dynamics in high-dimensional models. Indeed, **the single-index teacher has been widely adopted in the feature learning literature**, for example [1,2,3], since its design is essential and important for representing feature learning. Furthermore, we view our results with multiple time-scales as a first step toward understanding feature learning in more complex settings.
> > >
> > > [1] J. Ba et al. High-dimensional asymptotics of feature learning: how one gradient step improves the representation. NeurIPS (2022)
> > >
> > > [2] R. Berthier et al. Learning time-scales in two-layer neural networks. Foundations of Computational Mathematics (2025)
> > >
> > > [3] A. Montanari and P. Urbani. Dynamical decoupling of generalization and overfitting in large two-layer networks. NeurIPS (2025)
> > >
> > > ---
> > >
> > > Thank you for all your important questions. If you still have any questions, please feel free to ask.

---

### Decision · Program_Chairs · 2026-04-30

**Decision:**

Accept (regular)

**Comment:**

# Summary

This paper studies the optimization dynamics of a two-layer neural network trained with online minibatch SGD on single-index model data. By using a symmetric initialization and taking the mean-field limit, the paper derives a two-variable ODE that governs the dynamics. The authors observe that the ODE exhibits fast-slow dynamics: the alignment $R$ of first-layer weights develops rapidly, while the second-layer weight $a$ moves slowly. By studying the direction of the slow flow on a critical manifold (on which the fast dynamics converge), the paper numerically and theoretically demonstrates that feature unlearning, in which the feature is learned in the early phase and then is forgotten in the later stage, can happen in this model under certain conditions. The phenomenon is also observed in more realistic experiments.

# Comments

The concerns raised by reviewers were well addressed by the authors, resulting in mostly positive reviews after the discussion phase. I believe that the paper is a solid contribution that theoretically analyzes an interesting phenomenon of feature unlearning, so I recommend acceptance.

I would like to encourage the authors to carefully incorporate the points raised during the discussion, including comparisons with the existing literature (Montanari & Urbani in particular), presentation in Section 3.1 regarding the mean-field limit, and directly analyzing the ODE without the ansatz.

In addition to the points raised in the reviews, I would like to recommend that the authors address the following issues/questions in the final revision:
* In Section 3.2.2,  the notation around the fast/slow eigenvalues seems imprecise. In~(6), $f$ is scalar-valued, so $\nabla f(R_\tau,a_\tau)$ is a gradient rather than a Jacobian whose eigenvalues are defined. Presumably, the authors mean the Jacobian of the vector field $(f,g)$. Also, the statement $\lambda_f(\tau), \lambda_s(\tau) \in \mathbb{R}$ seems to require justification because the Jacobian is not necessarily symmetric.
* In the abstract, “(ii) an initial scale of the second-layer weights mitigates the feature unlearning” does not clearly convey how the weights should be scaled (larger or smaller) to mitigate unlearning.
* How much do your results depend on normalizing the $w_i$’s after each SGD update?
* The font for subsubsection titles looks slightly different from the template, although I’m not 100% sure.